



# Combining atmospheric and snow layer radiative transfer models to assess the solar radiative effects of black carbon in the Arctic

Tobias Donth[1], Evelyn Jäkel[1], André Ehrlich[1], Bernd Heinold[2], Jacob Schacht[2], Andreas Herber[3], Marco Zanatta[3], and Manfred Wendisch[1]

[1]Leipzig Institute for Meteorology (LIM), University of Leipzig, Germany
[2]Leibniz Institute for Tropospheric Research, (TROPOS), Leipzig, Germany
[3]Alfred Wegener Institute Helmholtz Centre for Polar and Marine Research (AWI), Bremerhaven, Germany

**Correspondence:** Evelyn Jäkel (e.jaekel@uni-leipzig.de)

**Abstract.** Solar radiative effects (cooling or warming) of black carbon (BC) particles suspended in the Arctic atmosphere and surface snow layer were explored by radiative transfer simulations on the basis of BC mass concentrations measured in pristine early summer and polluted early spring conditions under cloudless and cloudy conditions. To account for the radiative interactions between the black carbon containing snow surface layer and the atmosphere, a snow layer and an atmospheric

radiative transfer model were coupled iteratively. For pristine summer conditions (no atmospheric BC) and a representative BC particle mass concentration of $5\,\mathrm{ng\,g^{-1}}$ in the surface snow layer, a positive solar radiative effect of $+0.2\,\mathrm{W\,m^{-2}}$ was calculated for the surface radiative budget. Contrarily, a higher load of atmospheric BC representing springtime conditions, results in a slightly negative radiative effect of about $-0.05\,\mathrm{W\,m^{-2}}$, even when the same BC mass concentration is suspended in the surface snow layer. This counteracting of atmospheric BC and BC suspended in the snow layer strongly depends on the

snow optical properties determined by the snow specific surface area. However, it was found, that the atmospheric heating rate by water vapor or clouds is one to two orders of magnitude larger than that by atmospheric BC. Similarly, the total heating rate ($6\,\mathrm{K\,day^{-1}}$) within a snow pack due to absorption by the ice water, was found to be more than one order of magnitude larger than the heating rate of suspended BC ($0.2\,\mathrm{K\,day^{-1}}$). The role of clouds in the estimation of the combined direct radiative BC effect (BC in snow and in atmosphere) was analyzed for the pristine early summer and the polluted early spring BC conditions.

Both, the cooling effect by atmospheric BC, as well as the warming effect by BC suspended in snow are reduced in the presence of clouds.

## 1 Introduction

Black carbon (BC) aerosol particles originate from incomplete combustion of organic material (Bond et al., 2013). They strongly absorbs and scatters solar radiation in the visible wavelength range and, therefore, influence the Arctic solar radiative

energy budget. The manifold sources of BC particles and their atmospheric transport paths are well known (Law et al., 2014). However, the source strengths of the emissions are hard to quantify, which makes it challenging to reproduce the transport of BC particles into the Arctic (Stohl et al., 2013; Arnold et al., 2016; Schacht et al., 2019). Major sources of BC particles are forest fires, industry, and traffic predominantly located in lower latitudes; northern parts of Europe and America as well





as Siberia. Long-range transport in higher altitudes brings these emitted BC particles into the Arctic, where they can stay for several days (Liu et al., 2011). Contrarily, particles locally produced through ship traffic emissions and flaring from the oil industry settle down quickly on the surface and may alter the radiation budget within the snow pack (Bond et al., 2013). Nowadays local sources are only a minor component. Nevertheless, a strong intensification of the ship traffic is expected in the

future (Corbett et al., 2010). Still, the direct radiative impact by these additional BC particle emmissions is assumed to be of minor importance (Gilgen et al., 2018).

The BC particle mass concentration (in units $ng\,m^{-3}$) of suspended in the atmosphere is highly variable depending on the season and general meteorological conditions. In the case of BC particle plumes reaching the Arctic by long-range transport, atmospheric concentrations up to $150\,ng\,m^{-3}$ can be expected (Schulz et al., 2019). Sharma et al. (2013) compared atmospheric

BC particle mass concentrations measured during different Arctic campaigns. They identified large differences depending on region and season. Measurements in spring 2008 covering Alaska and northern Canada, showed values above $200\,ng\,m^{-3}$ in higher altitudes, while in spring 2009 more pristine air masses were encountered. In this period, the Arctic-wide airborne measurements indicated BC particle mass concentrations of less than $100\,ng\,m^{-3}$ in the entire vertical column.

To quantify the the amount of BC particles in a snow pack volume, the BC mass fraction (in units of $ng\,g^{-1}$) is used. Typical

values observed in Greenland range between 1 and $10\,ng\,g^{-1}$, in the Canadian Arctic between 5 and $20\,ng\,g^{-1}$, and in the northern parts of Russia values may reach $100\,ng\,g^{-1}$. Table 1 summarizes observational data of typical BC mass fractions in snow for different Arctic regions as reported by Doherty et al. (2010), Forsström et al. (2013), and Pedersen et al. (2015). Due to their absorbing effect, BC particles may contribute to the currently ongoing drastic Arctic climate changes, namely the Arctic Amplification (Wendisch et al., 2017). They can directly add to the warming of the atmosphere when suspended

in the air or to the reduction of the snow surface albedo if the BC is sedimented on the snow pack associated with a higher amount of absorbed radiation within the snow layer (Sand et al., 2013). Exemplarily, Warren (2013) estimated a decrease of 2 % in snow albedo in the visible spectral range for a snow pack with a BC mass fraction of $34\,ng\,g^{-1}$, which corresponds to the maximum value observed on the Greenland ice sheet (Doherty et al., 2010). More typical BC mass fractions in Arctic snow range between 5 and $20\,ng\,g^{-1}$ (Tab. 1), which would lead to a reduction of the snow surface albedo of around 1 %. For

typical Arctic summer conditions with a downward irradiance of $400\,W\,m^{-2}$ at the surface, a snow surface albedo reduction by one percent would lead to an additional absorption of solar radiative energy of $4\,W\,m^{-2}$ (Flanner et al., 2007). The additional absorption by BC supports the melting of snow and increases the snow grain size due to an enhanced snow metamorphism, which may lead to a further reduction of the surface albedo and, as a consequence, even more incoming solar radiation being absorbed. This positive feedback represents a self-amplifying process due to absorption by BC particles in snow. So far the

relevance of this feedback was not quantified.

BC particles suspended in the atmosphere, are known to influence the absorption and scattering of the incoming solar radiation. If atmospheric BC particles are located in high altitudes, enhanced backscattering and absorption of incoming solar radiation by the BC layer leads to a reduction of the solar radiation reaching the surface. At the same time, the absorbed radiation will warming the atmospheric BC layer. In extreme cases, the presence of atmospheric BC can effect the atmospheric

stability (Wendisch et al., 2008). The radiative heating of the lofted BC layers and the local cooling of the surface may enhance





**Table 1.** Typical values of the BC particle mass fraction in snow pack observed in different regions and seasons in the Arctic. Note, that Pedersen et al. (2015) derived the mass fraction of elemental carbon.

| Location | Season | BC mass fraction (ng g$^{-1}$) | Source |
|----------|--------|-------------------------------|--------|
| Svalbard region | March/April | 13 | Doherty et al. (2010) |
| Arctic Ocean snow | Spring | 7 | Doherty et al. (2010) |
| Arctic Ocean snow | Summer | 8 | Doherty et al. (2010) |
| Northern Norway | May | 21 | Doherty et al. (2010) |
| Central Greenland | Summer | 3 | Doherty et al. (2010) |
| Svalbard region | March/April | 11 - 14 | Forsström et al. (2013) |
| Corbel, Ny-Ålesund | March | 21 | Pedersen et al. (2015) |
| Barrow | April | 5 | Pedersen et al. (2015) |
| Ramfjorden, Tromsø | April | 13 | Pedersen et al. (2015) |
| Valhall, Tromsø | April | 137 | Pedersen et al. (2015) |
| Fram Strait | April | 22 | Pedersen et al. (2015) |

the already strongly stratified Arctic boundary layer over the snow and ice-covered areas, such that the atmospheric stability increases (Flanner, 2013).

   In general, the radiative effects of atmospheric BC particles and BC suspended in snow shows an opposite behavior (Flanner et al., 2007; Flanner, 2013; Sand et al., 2013). Model estimates of how these two effects balance each other, rely on the

accuracy of the assumed distribution of the BC particles. Samset et al. (2014) compared 13 aerosol models from the AeroCom Phase II; all of them included BC as an aerosol species. They found that modeled atmospheric BC concentrations often show a spread over more than one order of magnitude. In remote regions, dominated by long range transport, these models tend to overestimate the atmospheric BC concentrations compared to airborne observations. On the other hand, an underestimation of deposition rates induces a lower BC mass fraction in snow (Namazi et al., 2015). However, long-term trends and mean

multi-model results were representative for Arctic-wide observations (Sand et al., 2017).

   Most previous studies quantifying the radiative impact of BC particles either focused on estimates of cooling/heating effects in the atmosphere (e.g., Samset et al., 2013) or on radiative effects of BC in the snow surface layer (Dou and Cun-De, 2016, and references therein). In contrast, this paper will combine both effects by iteratively coupling radiative transfer simulations in both compartments, the atmosphere and the snow pack. For typical Arctic BC distributions and concentrations for spring and

summer months, the local direct radiative effects of BC particles suspended in the snow surface layer and in the atmosphere are quantified. with this approach, the interactions between the BC effects in the atmosphere and the snow pack will be considered. In particular, the role of clouds on the cooling/heating effect caused by BC particles will be examined. Due to the fact that clouds enhance the atmospheric multi-scattering between surface and cloud layer, but also enhance the surface albedo (Choudhury and Chang, 1981), it is expected that clouds alter also the radiative impact by BC particles. To our knowledge, this interaction

was not explicitly discussed in previous publications.



The radiative transfer simulations used in this study are based on airborne observations of atmospheric BC concentration in the Arctic, which were taken during three field campaigns in the European and Canadian Arctic. The applied models and observations are introduced in Section 2. Section 3 discusses the radiative effects of BC particles on the surface solar radiative budget. Vertical profiles of heating rates induced by the atmospheric BC particles and BC particles in the snow pack are

presented. To estimate the relevance of BC particles, effective heating rates are calculated by separating the pure BC effect from the total heating rates.

## 2 Setup of radiative transfer simulations and model coupling

### 2.1 Aircraft campaigns and BC data

The atmospheric model setup was adapted to campaign specific conditions. Measured profiles of the atmospheric BC taken

from three aircraft campaigns were taken into account, which represent typical cases with higher BC concentrations (polluted case) in spring with low sun, and lower BC concentration (pristine conditions) in summer during the polar day. The atmospheric BC particle mass concentrations were derived from airborne measurements with a Single Particle Soot Photometer (SP2, Moteki and Kondo, 2007). The Arctic Research of the Composition of the Troposphere from Aircraft and Satellites (ARCTAS) spring campaign was performed in April 2008 (Jacob et al., 2010; Matsui et al., 2011). The aircraft operation of ARCTAS

mainly took place in northern Alaska and the Arctic Ocean. Similar SP2 measurements were performed during the Polar Airborne Measurements and Arctic Regional Climate Model Simulation Project (PAMARCMiP) campaigns which is a series of aircraft observations performed within the Arctic region (Herber et al., 2012; Stone et al., 2010). Here data from PAMARCMiP 2018 are analysed which was based at the Villum Research Station (Station Nord/Greenland) and conducted flights from 10 March to 8 April 2018 in the European Arctic. In contrast to both spring campaigns, the Arctic CLoud Observations

Using airborne measurements during polar Day (ACLOUD) campaign was conducted in early summer 2017 characterizing the atmosphere over the Arctic Ocean north and west of Svalbard (Wendisch et al., 2019). ACLOUD was coordinated with the Physical Feedbacks of Arctic Boundary Layer, Sea Ice, Cloud and Aerosol (PASCAL) cruise of the research vessel Polarstern which provides a ground-based characterization of snow properties (Macke and Flores, 2018). Besides the differences in atmospheric BC concentrations, also the range of the daily solar zenith angle (SZA) and, therefore, the available incoming

solar radiation, varied significantly for the three campaign periods. When analysing the radiative impact of BC on basis of daily averages, the magnitude of solar incident radiation and the length of the day play a major role. While the summer conditions of ACLOUD are characterized by the polar day and SZA between 55° and 78°, during ARCTAS the available incoming solar radiation was lower due to a nighttime of about 8.5 hours and a minimum SZA of 62.5°. PAMARCMiP was conducted in the most northern region and earlier in the year, such that the Sun was about 9.5 hours below the horizon and the minimum SZA

was 79° at noon. Table 2 summarizes the key characteristics of the three analysed data sets.

Campaign specific BC particle mass concentration profiles were calculated by averaging over all available aircraft measurements. Figure 1 shows the different BC profiles composed in discrete layers as implemented into the radiative transfer simulations. The ACLOUD case represents nearly pristine BC conditions with maximum values of up to $12\,\mathrm{ng\,m^{-3}}$, while





**Table 2.** Region, period, solar zenith angle range, and maximum BC particle mass concentration and mean optical depth of BC at 500 nm wavelength characterizing the three data sets obtained within ARCTAS, ACLOUD, and PAMARCMiP.

|  | ARCTAS | ACLOUD | PAMARCMiP |
|---|---|---|---|
| Region | Alaska/ Northern Canada | Svalbard/Arctic Ocean | Northern Greenland/ Arctic Ocean |
| Latitude (°) | 71 | 78 | 82 |
| Period | April 2008 | May/June 2017 | March/April 2018 |
| SZA (°) | 63–90 | 55–78 | 79–90 |
| Night length (h) | 8.6 | 0.0 | 9.4 |
| Max. BC concentration (ng m$^{-3}$) | 149 | 13 | 117 |
| BC optical depth at 500 nm | 0.008 | 0.0003 | 0.006 |
| Reference | Jacob et al. (2010) | Ehrlich et al. (2019) | Herber (2019) |

the ARCTAS and PAMARCMiP profiles show 10-20 times larger BC concentration. Due to the different flight performances of the aircrafts (ARCTAS used a DC-8 while ACLOUD/PAMARCMiP operated the Polar 5 and 6, two modified DC-3 aircraft of the type Basler BT-67), the profiles measured in the European Arctic are restricted to 5.5 km altitude (Jacob et al., 2010; Ehrlich et al., 2019; Herber, 2019). However, on the Polar 5 aircraft Sun photometer are available. The observations

5   during ACLOUD indicate, that the extinction above the maximum flight altitude of 5.5 km does was negligible. During PAMARCMiP, additionally the upward-looking Airborne Mobile Aerosol Lidar for Arctic research (AMALi) was installed on the aircraft (Stachlewska et al., 2010). Based on the backscattering signal potential aerosol layer above the maximum flight level were observed only in one of the 14 flights.Therefore, the constructed profiles of PAMARCMiP and ACLOUD are assumed to be representative for Arctic spring and summer conditions, respectively.

## 2.2   Atmospheric radiative transfer model

To simulate vertical profiles of the spectral upward and downward irradiance the library for radiative transfer routines and programs (libRadtran, Emde et al., 2016; Mayer and Kylling, 2005) was used. The model also provides the ratio of the direct-to-global irradiance $f_{\mathrm{dir/glo}}$, which is required as a boundary condition of the snow pack radiative transfer model. As a solver

15   for the radiative transfer equation, the Discrete Ordinate Radiative Transfer solver (DISORT) 2 (Evans, 1998; Stamnes et al., 2000) routine was chosen from the libRadtran package. The spectral resolution of the simulations was set to 1 nm covering a wavelength range between 350 nm and 2400 nm. The extraterrestrial spectrum was taken from the solar spectrum from Gueymard (2004).

    The meteorological input for the model is based on standard profiles of trace gases, temperature, and pressure from Anderson

20   et al. (1986). Sub-Arctic summer conditions were chosen for the summer case and subarctic winter conditions for the winter and early spring cases. The standard profiles were modified by observations from radio soundings near the research area or





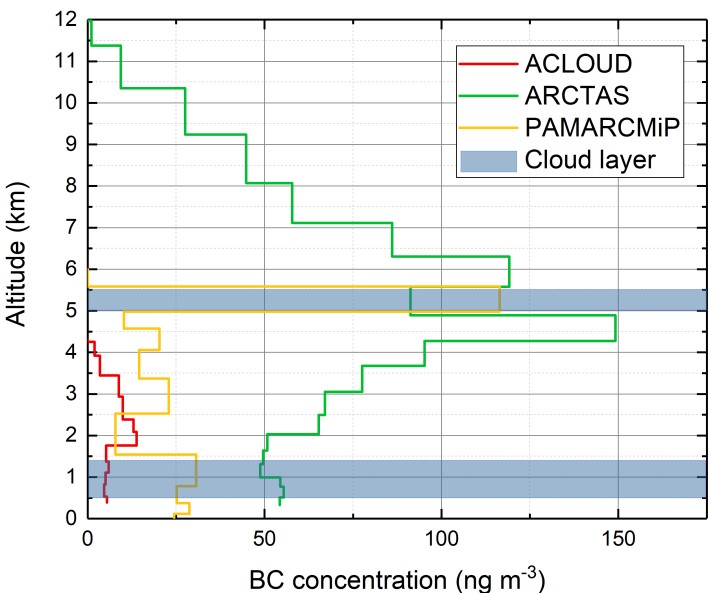

**Figure 1.** Campaign specific mean BC profiles as used for the radiative transfer simulations. Additional the positions of two assumed cloud layers (blue shaded area) are marked.

dropsondes released during the flights. The range of the SZA and the atmospheric profiles were adjusted to represent the middle of the individual campaign periods.

To account for the impact of the campaign specific atmospheric BC profiles (Fig. 1), the BC optical properties including the refractive index, density, extinction coefficient, single scattering albedo, and scattering phase function from the OPAC

5 aerosol database were applied (Hess et al., 1998). To test the sensitivity of the BC radiative effects on cloud occurrence, two cloud layers were implemented in the atmospheric profiles as illustrated in Fig. 1. The cloud layers were constructed to represent typical Arctic cloud conditions. A low-level liquid water cloud was placed between 500 m and 1.4 km according to observations of Arctic clouds presented by Bierwirth et al. (2013) and Leaitch et al. (2016). The liquid water content increases from $0.1\,\mathrm{g\,m^{-3}}$ at cloud base to $0.3\,\mathrm{g\,m^{-3}}$ at cloud top, the cloud particle effective radius increased from 6 $\mu$m to 12 $\mu$m. The

10 second cloud layer represents a thin ice water cloud and was positioned between 5 and 5.5 km. This thin cloud was assumed to be homogeneous with an ice water content of $0.006\,\mathrm{g\,m^{-3}}$ and an effective cloud particle radius of 40 $\mu$m, according to airborne measurements reported by Wyser (1998) and Luebke et al. (2013).





### 2.3 Snow pack radiative transfer model

The Two-streAm Radiative TransfEr in Snow model (TARTES) was used to simulate the radiative transfer in the snow pack (Libois et al., 2013, 2014). In TARTES, the snow profile can be constructed of a predefined number of horizontally homogeneous snow layers which allows to account for the stratifiction of the snow pack. To account for the single scattering properties of each layer, TARTES applies the method described by Kokhanovsky and Zege (2004). To solve the radiative transfer equation, the delta-Eddington approximation (Stamnes et al., 1988) is applied. The model computes the spectral surface albedo and the profile of the irradiance within the snow pack. As boundary condition the SZA and $f_{\mathrm{dir/glo}}$ have to be defined. For each of the snow layers the optical and microphysical properties, such as the snow density ($\rho_{\mathrm{ice}}$), the specific surface area ($SSA$), and the snow grain shape parameter, which represents a mixture of different grains as suggested by Libois et al. (2013), were set. The specific surface area can be translated into the optical snow grain size $r_{\mathrm{opt}}$ by:

$$r_{\mathrm{opt}} = \frac{3}{\rho_{\mathrm{ice}} \cdot SSA}, \tag{1}$$

TARTES allows to add impurities ot each snow layer characterized by the impurity type and mass fraction. Note, that the impurities considered in TARTES are assumed to be externally mixed. To simulate a BC-containing snow layer, the complex refractive index and the density of BC particles given by Bond et al. (2013) were applied.

The input parameters of the snow pack model are summarized in Table 3. For the bottom layer, a soil albedo of 0.3 was assumed representing the conditions below the snow pack. The impact of the soil albedo on the albedo of the snow surface depends on the depth of the overlying snow pack. Sensitivity studies have shown, that for snow depth of more than 20 cm the albedo of a snow surface is independent of the choice of the soil albedo below. In this study the snow pack depth was set to 1 m thickness. Reference simulations of a pristine snow layer were performed by assuming only one single homogeneous snow layer. Simulations including BC impurities were based on BC particle mass fractions measured by several studies reported in literature as summarized in Table 1 (Doherty et al., 2010; Forsström et al., 2013; Pedersen et al., 2015) and observations during PASCAL and PAMARCMiP. For the simulations of a single homogeneous snow layer, typical BC particle mass fractions of 5 ng g$^{-1}$ and 20 ng g$^{-1}$ were chosen. The default snow density and $SSA$ were set to 300 kg m$^{-3}$ and 20 m$^2$ kg$^{-1}$, respectively. To analyze the sensitivity of the snow surface albedo with respect to the snow grain size, $SSA$ values of 5 m$^2$ kg$^{-1}$ and 60 m$^2$ kg$^{-1}$ were used.

In addition to the simulations of a homogeneously mixed snow layer, a second model setup used in Section 3.2.2 considers a multi-layer snow pack. Based on snow pit measurements in Greenland, Doherty et al. (2010) identified typical multi-layer structures, where BC accumulated in a melting layer approximately 10 cm below the surface. Referring to these measurements, the snow pack of the second model setup consists of three snow layers. The top layer is 5 cm and the BC-containing middle layer is 10 cm thick. The bottom layer below continues to 1 m depth. For this multi-layer approach, BC was mainly included in the middle layer, representing an aged melting layer in which, impurities have accumulated ($SSA = 20$ m$^2$ kg$^{-1}$, snow density of 350 kg m$^{-3}$, and a BC mass fraction of 15 ng g$^{-1}$). The top layer is assumed to be of fresh and clean snow with $SSA = 40$ m$^2$ kg$^{-1}$, a snow density of 250 kg m$^{-3}$, and a BC mass fraction of 2 ng g$^{-1}$). The aged snow layer at bottom was





characterized by an enhanced snow grain size and density of $SSA = 10\,\mathrm{m^2\,kg^{-1}}$ and $\rho_{\mathrm{ice}} = 450\,\mathrm{kg\,m^{-3}}$, respectively, and a BC mass fraction of $2\,\mathrm{ng\,g^{-1}}$.

**Table 3.** Snow pack model setups for the single layer and multi-layer cases. The default $SSA$ for the single layer case is $20\,\mathrm{m^2\,kg^{-1}}$.

|  | Single layer | Multi-layer | | |
|---|---|---|---|---|
|  |  | top layer | middle layer | bottom layer |
| Depth (cm) | 100 | 5 | 10 | 85 |
| BC mass fraction ( $\mathrm{ng\,g^{-1}}$ ) | 5 / 20 | 2 | 15 | 2 |
| $SSA$ ($\mathrm{m^2\,kg^{-1}}$) | 5 / 20 / 60 | 40 | 20 | 10 |
| Density ($\mathrm{kg\,m^{-3}}$) | 300 | 250 | 350 | 450 |

## 2.4 Iterative coupling

The surface albedo is an important boundary condition to simulate the radiative transfer in the atmosphere. At the same time,

the spectral surface albedo depends on the illumination conditions defined by the solar zenith angle, the spectral distribution of downward irradiance, and the ratio of direct-to-global irradiance (e.g., Wiscombe and Warren, 1980; Gardner and Sharp, 2010). Especially a transition from cloudy to cloudless conditions significantly increases the direct-to-global ratio ($f_{\mathrm{dir/glo}}$) and shifts the downward irradiance to shorter wavelengths (Warren, 1982). Therefore, a cloud cover typically increases the broadband surface albedo. Exemplarily, comparing cloudless and cloudy conditions with TARTES lead to a change of the broadband

snow surface albedo from about 0.8 to 0.9 for a SZA of 60°. As clouds mostly absorb solar radiation at wavelengths larger than 1000 nm, the shorter wavelengths where BC particles absorb solar radiation become more relevant. However, it is not sufficient to consider only the two cases of pure cloudless and cloudy conditions. For optically thin clouds or the presence of atmospheric pollution layers, $f_{\mathrm{dir/glo}}$ can quickly change with slight changes of the cloud and atmosphere properties (e.g., liquid water path or aerosol optical depth). To account for the variability of these effects, the atmospheric and the snow pack radiative transfer

models need to be coupled. Therefore, an iterative method coupling libRadtran and TARTES via their boundary conditions, surface albedo and direct-to-global ratio ($f_{\mathrm{dir/glo}}$) of the incident radiation, was applied. Both parameters were transferred between the models as schematically illustrated in Fig. 2. In the first iteration step, only diffuse radiation was assumed ($f_{\mathrm{dir/glo}}$ = 0) to calculate the snow surface albedo by TARTES, which serves as input for the libRadtran simulations. Then a new spectral direct-to-global ratio representing the atmospheric conditions was calculated by libRadtran, which is in turn used to re-adjust

TARTES, starting a revised iteration (n+1) to calculate a new spectral surface albedo $\alpha_\lambda$(n+1). This procedure is repeated until the deviation between previous (step n) and revised surface albedo decrease below 1 %. Exemplarily, Figure 3 illustrates the change of the spectral surface albedo for a cloudless case without atmospheric BC and a SZA of 60°. The BC mass fraction in snow was set to $5\,\mathrm{ng\,g^{-1}}$. Two iteration steps were necessary to match the termination criteria, which was a typical number for all studied cases. Starting with pure diffuse conditions allows faster calculations in particular for cloudy cases. This quick


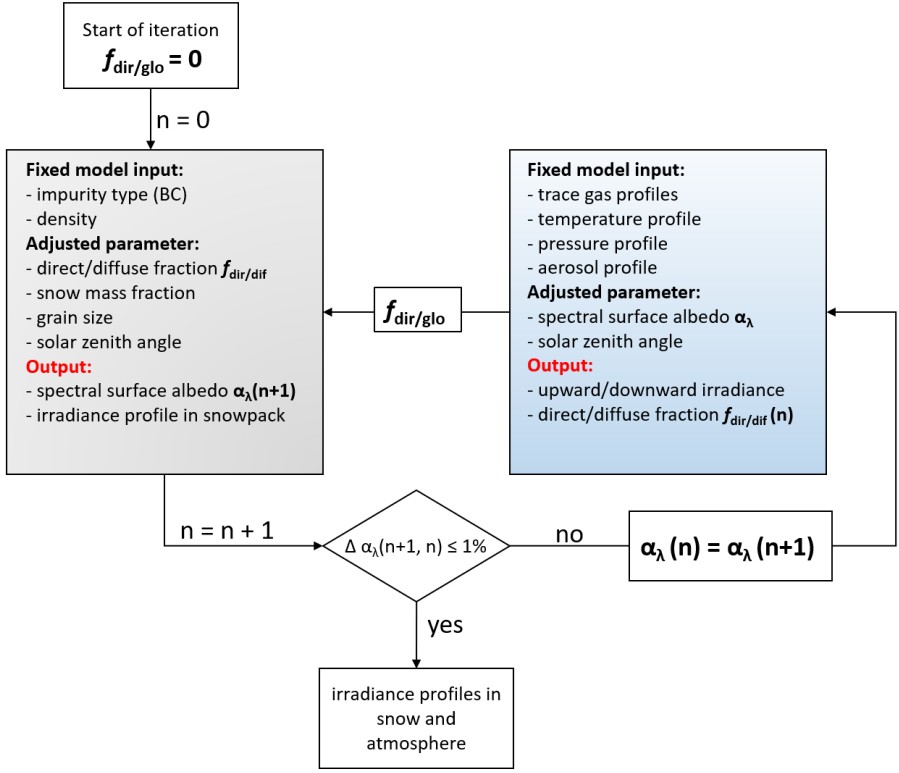

**Figure 2.** Schematics of the coupling of TARTES (gray box) and libRadtran (blue box) by exchanging the spectral surface albedo and the direct-to-global ratio. The major model input parameters are listed in the boxes.

conversion of the iteration allows considering different cloud properties and atmospheric conditions and facilitates to calculate the radiative effects of BC particles in the atmosphere and within the snow pack simultaneously.

## 2.5 Quantities used to characterize the impact of BC particles

In this study, the surface radiative effect of BC particles and profiles of heating rates were analyzed. The total radiative effect at
5  the surface $\Delta F_{\mathrm{tot}}$ is separated into the effect of BC particles suspended in the atmosphere $\Delta F_{\mathrm{atm}}$ and the effect of BC particles deposited in the snow pack $\Delta F_{\mathrm{snow}}$. The separated effect of BC suspended in the snow $\Delta F_{\mathrm{snow}}$ is defined by the difference of the net irradiance (difference of downward and upward solar irradiance) of the case including BC in the snow layer $F_{\mathrm{net,BC}}$ and the clean reference case without BC in the snow layer $F_{\mathrm{net,clean}}$. Similarly, $\Delta F_{\mathrm{atm}}$ is defined as the difference between the net irradiances derived for BC in snow and atmosphere and the BC-free reference case:

10  $$\Delta F_{\mathrm{tot/atm/snow}} = F_{\mathrm{net,BC}} - F_{\mathrm{net,clean}}. \tag{2}$$

For the separated effects, $F_{\mathrm{net,clean}}$ refers to either a clean atmosphere or a clean snow layer, while the other part can contain BC particles. For $\Delta F_{\mathrm{tot}}$, the clean reference assumes both a pristine atmosphere and pristine snow layer.



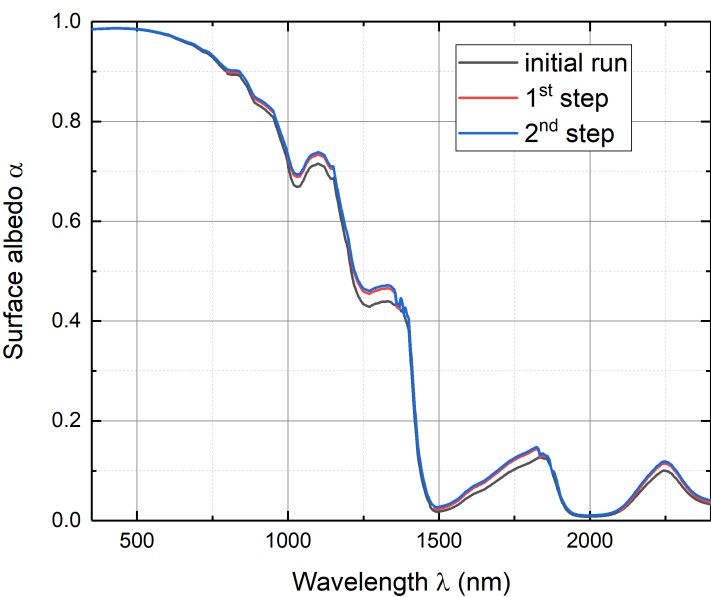

**Figure 3.** Change of the spectral snow albedo for cloudless conditions with a SZA of $60°$ due to the iterative adjustment by the coupled atmosphere and snow radiative transfer models. The initial run assumes a direct-to-diffuse ratio of zero.

The calculation of atmospheric and snow heating rate profiles $HR(z)$ (in $\mathrm{K\,day^{-1}}$) is based on the net irradiances at the top (t) and bottom (b) of selected atmospheric or snow layer $z$, the layer density $\rho(z)$, the specific heat capacity under constant pressure $c_\mathrm{p}$, and the layer thickness $(z_\mathrm{t} - z_\mathrm{b})$:

$$HR(z) = \frac{\Delta T}{\Delta t}(z) = \frac{F_\mathrm{net}(z_\mathrm{t}) - F_\mathrm{net}(z_\mathrm{b})}{\rho(z) \cdot c_\mathrm{p} \cdot (z_\mathrm{t} - z_\mathrm{b})} \qquad . \tag{3}$$

5   For atmospheric profiles, $c_\mathrm{p,air} = 1004\,\mathrm{J\,kg^{-1}\,K^{-1}}$ the vertical resolution of the simulated irradiances was adjusted to the vertical resolution of the measured BC profiles, ranging between 500 m and 1 km. Increasing the vertical resolution has shown only negligible differences of less than 1 %. Similarly, the heating rate profiles within the snow pack were calculate with Eq. 3 by accounting for the snow density (set to $300\,\mathrm{kg\,m^{-3}}$) and the specific heat capacity of ice $c_\mathrm{p,snow} = 2060\,\mathrm{J\,kg^{-1}\,K^{-1}}$ at a temperature of $0\,°\mathrm{C}$. The layer thickness within TARTES and therefore, the resolution of the heating rate profiles was of 1 cm.

10   To separate the contribution of BC particles to the total heating rate, the effective BC heating rate $HR_\mathrm{BC}(z)$ was calculated as the difference between the total heating rate $HR_\mathrm{tot}(z)$ and the heating rate of the clean reference case $HR_\mathrm{clean}(z)$:

$$HR_\mathrm{BC}(z) = HR_\mathrm{tot}(z) - HR_\mathrm{clean}(z). \tag{4}$$

If not indicated differently, radiative effects reported in this study refer to daily mean estimates accounting for the change of the SZA and the night time. Therefore, simulations were performed for a full diurnal cycle with a temporal resolution of





five minutes. The simulated upward and downward irradiance were averaged. Then these daily mean irradiances are applied to calculate mean values of $\Delta F$ and $HR(z)$.

## 3 Results

### 3.1 Radiative impact of BC at surface level

5    #### 3.1.1 Effect on surface albedo

The effect of BC impurities on the snow surface albedo are known to depend on the snow grain size ($SSA$ respectively). Here, changes of the snow surface albedo due to the combination of BC impurities and snow grain size variations are evaluated for Arctic conditions. The single-layer snow pack setup, as defined in Section 2, was used together with atmospheric properties representing the ACLOUD conditions. The SZA was set to a constant value of $60°$. Figure 4 shows the spectral snow albedo for variable BC particle mass fractions (0, 5, and $20\,\mathrm{ng\,g^{-1}}$). The selected $SSA$ values represent different snow types, as freshly fallen snow with small snow grains ($SSA = 60\,\mathrm{m^2\,kg^{-1}}$), aged snow which has undergone snow metamorphism ($SSA = 5\,\mathrm{m^2\,kg^{-1}}$) when surface temperature approaches $0°$C, and moderate aged snow without melting ($SSA = 20\,\mathrm{m^2\,kg^{-1}}$), which in this study is considered as default case. As expected, the highest values of surface albedo are obtained for the case with clean and fresh snow. Adding BC particles causes a decrease in the spectral surface albedo in particular in the visible spectral range up to 700 nm, shown in the enlargement of Fig. 4. In contrast, the near-infrared spectral range is dominated by ice absorption which is affected by the $SSA$ (grain size). From the simulations shown in Fig. 4 it becomes apparent that the decrease of surface albedo with increasing BC mass fraction is stronger for aged snow than for fresh snow. Fresh snow with smaller grains leads to an enhanced backscattering of the incoming radiation, while larger grains allow deeper penetration of the incident radiation into the snow pack. Since the penetration depth for aged snow is deeper, the probability is higher, that the radiation gets absorbed by the BC particles leading to a decrease of the spectral surface albedo.

In the same way, the radiative effect of BC particles suspended in the snow layer was calculated for overcast cloudy conditions (predefined low-level liquid water cloud case) in order to assess the relevance of changes of the BC mass fraction compared to variations in $SSA$ and the illumination conditions. To estimate the relevance for the surface energy budget, the solar broadband effect is analyzed by calculating the broadband albedo $\alpha_{\mathrm{bb}}$. Therefore, the spectral albedo simulated by TARTES and the spectral downward irradiance $F_\lambda^\downarrow(\lambda)$ simulated by libRadtran are used:

$$\alpha_{\mathrm{bb}} = \frac{\int \alpha(\lambda) \cdot F_\lambda^\downarrow(\lambda)\,d\lambda}{\int F_\lambda^\downarrow(\lambda)\,d\lambda}. \tag{5}$$

The calculated broadband albedo are summarized in Table 4 for the cloudy and cloudless case. For both cases, the highest BC mass concentration does reduce the surface albedo by less than 1 %. Contrarily, the snow grain size and the presence of clouds show significant changes of the snow albedo. The difference of the broadband surface albedo between fresh and aged snow ranges up to 0.12 and 0.08 for cloudless and cloudy conditions, respectively, which is in the same order than the effect of the presence of clouds (0.12 for fresh snow and 0.07 for aged snow). Therefore, the impact of BC particles suspended in the



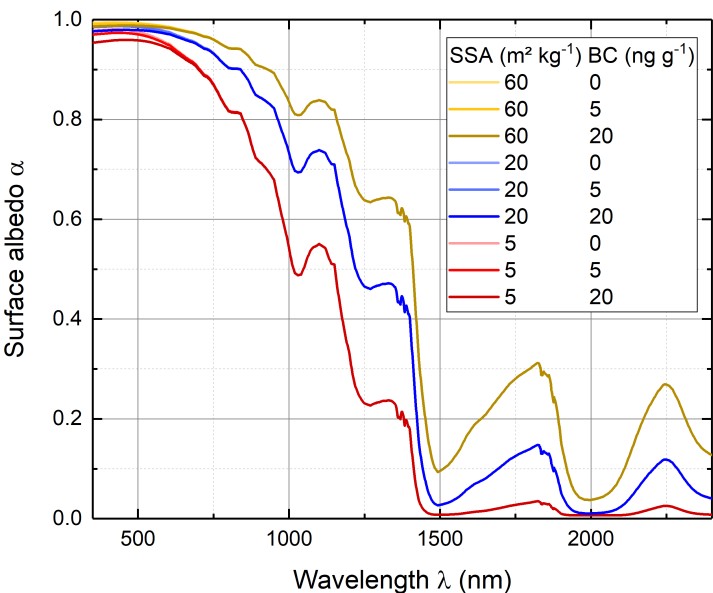

**Figure 4.** Spectral surface albedo of snow for cloudless conditions and a SZA of $60°$ for different $SSA$ and BC particle mass fractions.

snow pack is assumed to be of minor importance for Arctic conditions, which is in agreement with findings by e.g., Warren (2013).

**Table 4.** Broadband surface albedo ($\alpha_{bb}$) of fresh ($SSA = 60\,\mathrm{m^2\,kg^{-1}}$) and aged snow ($SSA = 5$ and $20\,\mathrm{m^2\,kg^{-1}}$) depending on the BC partcile mass concentration and illumination condition.

| $SSA$ (m² kg⁻¹) | Cloudless case $\alpha_{bb}$ BC mass fraction (ng g⁻¹) | | | Cloudy case $\alpha_{bb}$ BC mass fraction (ng g⁻¹) | | |
|---|---|---|---|---|---|---|
| | 0 | 5 | 20 | 0 | 5 | 20 |
| 5 | 0.76 | 0.76 | 0.75 | 0.88 | 0.87 | 0.87 |
| 20 | 0.83 | 0.83 | 0.82 | 0.92 | 0.92 | 0.92 |
| 60 | 0.87 | 0.87 | 0.87 | 0.95 | 0.95 | 0.94 |

### 3.1.2 Surface radiative effects

The decrease of the snow surface albedo due to the increase of BC particle mass concentration and snow grain size directly effects the surface radiative effect of the snow pack $\Delta F_{snow}$. Applying Eq. 2, $\Delta F_{snow}$ was calculated for a typical Arctic range of BC particle mass fractions in snow and $SSA$ values assuming the ACLOUD atmospheric conditions and a fixed solar zenith




angle of $60°$. Figure 5 shows a contour plot of $\Delta F_{\mathrm{snow}}$ for all combinations of $SSA$ and BC particle mass fractions. For a BC particle mass fraction of $5\,\mathrm{ng\,g^{-1}}$ representing rather clean conditions, $\Delta F_{\mathrm{snow}}$ ranges between $0.1-0.4\ \mathrm{W\,m^{-2}}$. Higher BC particle mass fractions increase $\Delta F_{\mathrm{snow}}$ depending on the snow grain size ($SSA$ respectively). The strongest increase of the solar radiative warming was calculated for small $SSA$ values, corresponding to larger snow grain sizes. With the larger

5   penetration depth for a smaller $SSA$, more radiation can be absorbed by the BC particles. For typical BC concentrations in the central Arctic, which are below $10\,\mathrm{ng\,g^{-1}}$, the maximum BC radiative effect is about $1\,\mathrm{W\,m^{-2}}$.

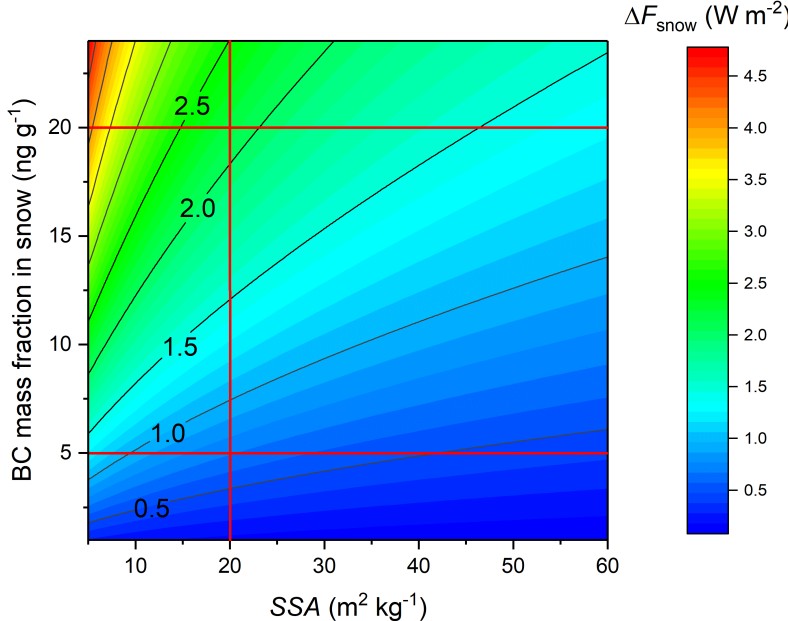

**Figure 5.** Solar surface radiative effects of BC impurities in snow $\Delta F_{\mathrm{snow}}$ calculated for different $SSA$ values and BC particle mass fractions. The atmospheric conditions correspond to the ACLOUD case with a fixed SZA of $60°$. Horizontal red lines indicate typical Arctic conditions with rather clean and more polluted snow; the vertical line represents the most typical $SSA$.

To compare the radiative effect of atmospheric BC particle profiles observed during the three aircraft campaigns ACLOUD, PAMARCMiP, and ARCTAS, daily averaged surface radiative effects are analyzed. To limit the degree of freedom, the $SSA$ was set to a default value of $SSA = 20\,\mathrm{m^2\,kg^{-1}}$ representative for snow covered Arctic sea ice. To estimate the relevance

10   of the atmospheric BC particles, their separated radiative effect $\Delta F_{\mathrm{atm}}$ was calculated. Additionally, the total radiative effect $\Delta F_{\mathrm{tot}}$ combining the atmospheric and snow BC was analyzed. Figure 6 summarizes the daily averaged $\Delta F_{\mathrm{snow}}$ (panel a), $\Delta F_{\mathrm{atm}}$ (panel b), and $\Delta F_{\mathrm{tot}}$ (panel c) for different BC particle mass fractions in snow (0, 5, 20 $\mathrm{ng\,g^{-1}}$) in cloudless and cloudy conditions.





The BC particles suspended in snow lead to warming effects of up to $0.9 \, \mathrm{W \, m^{-2}}$ for high BC mass fractions of $20 \, \mathrm{ng \, g^{-1}}$ and ACLOUD conditions. For ARCTAS $\Delta F_{\mathrm{snow}}$ is slightly lower and for PAMARCMIP reduced by a factor of about 3. This difference is caused by the lower maximum Sun elevation during PAMARCMiP (location in higher latitude) resulting in a lower amount of available incoming solar irradiance compared to ACLOUD and ARCTAS (see range of SZA in Tab. 2).

Atmospheric BC particles reduce the incident solar radiation at surface due to extinction, such that the atmospheric radiative effect $\Delta F_{\mathrm{atm}}$ is negative in all scenarios (Figure 6b). This cooling at the surface is strongest with values up to $-0.2 \, \mathrm{W \, m^{-2}}$ in cloudless conditions for the ARCTAS case, where the largest atmospheric BC particle concentrations were observed. Despite having a BC optical depth of similar magnitude, the PAMARCMiP case ($\mathrm{AOD_{BC}} = 0.006$) shows a weaker radiative cooling compared to the ARCTAS case ($\mathrm{AOD_{BC}} = 0.008$) caused by the higher solar zenith angles in PAMARCMiP. Minor cooling

of less than $-0.02 \, \mathrm{W \, m^{-2}}$ is observed for the ACLOUD case, where the atmosphere is rather clear with significant reduced atmospheric BC particle concentrations (factor ten lower than during ARCTAS). Comparing the simulations with different BC mass fractions in snow shows only little effects of the surface properties on the radiative effect of atmospheric BC. A slight decrease of $\Delta F_{\mathrm{atm}}$ with increasing BC mass fractions is observed for the ARCTAS case indicating, that a lower surface albedo enhances the radiative effects of atmospheric BC particles.

The cooling effect of atmospheric BC counteracts the warming effect of BC particles in snow and can lead to positive and negative total radiative effects. Figure 6c shows the total radiative effect $\Delta F_{\mathrm{tot}}$ for all cases. For BC mass fraction of $20 \, \mathrm{ng \, g^{-1}}$, all cases show a total warming effect when the warming of BC in the snow pack exceeds the cooling by atmospheric BC. The strongest warming effect of up to $0.7 \, \mathrm{W \, m^{-2}}$ is found for the ACLOUD case which is characterized by the pristine atmospheric conditions in the Arctic summertime. For less polluted snow ($5 \, \mathrm{ng \, g^{-1}}$), warming and cooling scenarios can occur depending

on the concentration of atmospheric BC (ARCTAS shows a slight cooling) and the solar zenith angle (ACLOUD shows a significant warming effect). $\Delta F_{\mathrm{tot}}$ calculated for ACLOUD even exceeds the warming effect of PAMARCMiP for the higher BC mass fraction in the snow layer. This clearly demonstrates that the competition between the individual BC radiative effects $\Delta F_{\mathrm{atm}}$ and $\Delta F_{\mathrm{snow}}$ is strongly driven by solar zenith angle and the available solar radiation and is less affected by the BC concentrations itself.

The available solar irradiance is strongly affected by the presences of clouds. Therefore, the impact of clouds on the BC radiative effects was analyzed. Two cloud layers as defined in Section 2.2 were implemented in the simulations and considered in the calculation of $\Delta F_{\mathrm{tot/atm/snow}}$ (clean cloudy and polluted cloudy case in Eq. 2) to extract the pure BC radiative effect. In Fig. 6 the BC radiative effects of the cloudy scenarios are shown by the shaded bars. The magnitudes of $\Delta F_{\mathrm{snow}}$ (panel a) and $\Delta F_{\mathrm{atm}}$ (panel b) are always reduced by the presence of clouds. $\Delta F_{\mathrm{snow}}$ drops by about 15 % in all cases ($0.1 \, \mathrm{W \, m^{-2}}$

for ALCOUD and ARCTAS and high BC mass concentration in snow), while $\Delta F_{\mathrm{atm}}$ increases by more than 50 %. $\mathrm{W \, m^{-2}}$, which amounts for ARCTAS to an absolute increase of $0.14 \, \mathrm{W \, m^{-2}}$. Cloud reduce $\Delta F_{\mathrm{snow}}$ because less radiation reaches the surface and can be absorbed by BC particles in the snow pack. The shift from a mostly direct illumination of the snow surface by the Sun to a diffuse illumination below the clouds is less significant as demonstrated in Table 4.

     These different cloud effects counterbalance in the total radiative effects $\Delta F_{\mathrm{atm}}$ (Fig. 6c). To illustrate the total effect by

clouds, Fig. 6d shows the difference between cloudy and cloudless simulations. In all scenarios, still slight differences between





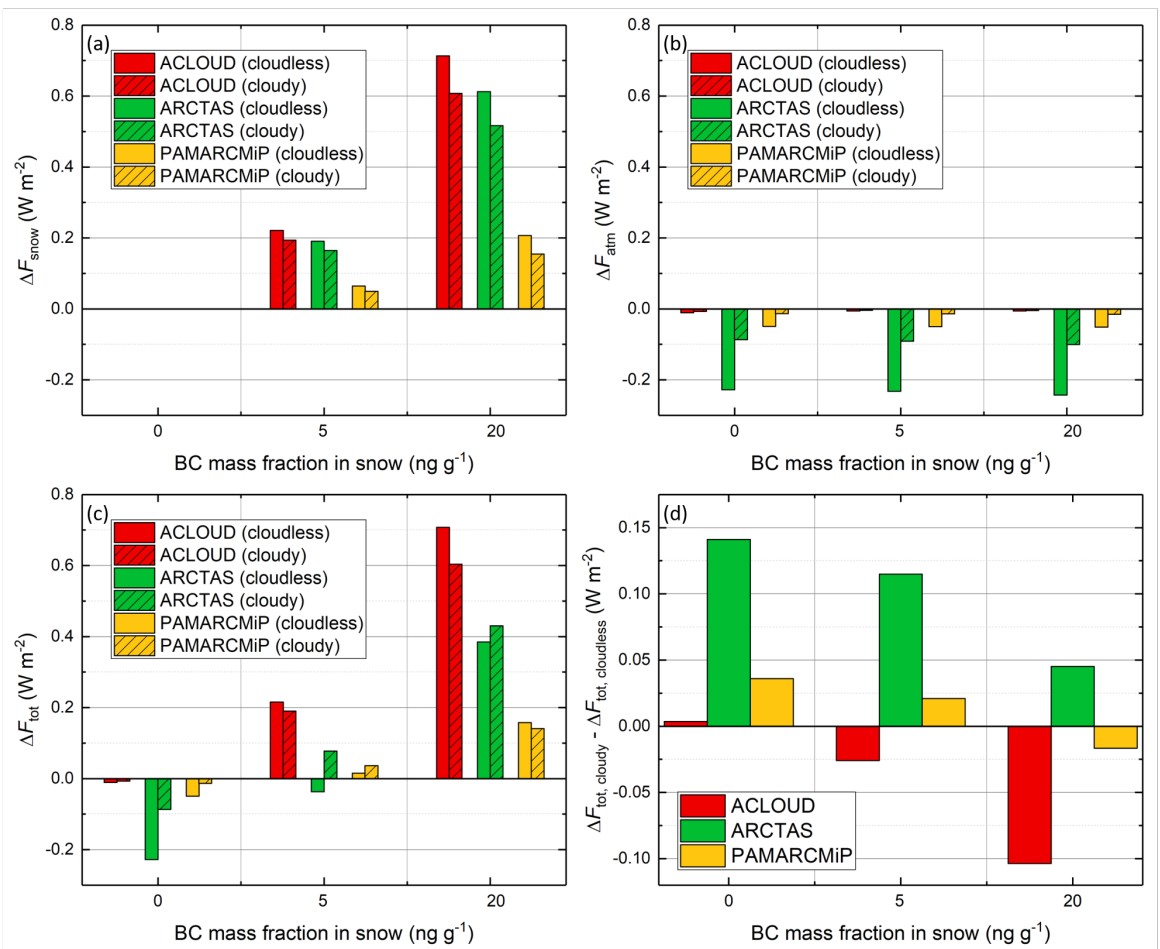

**Figure 6.** Daily mean of the solar surface radiative effects simulated for the conditions of the three campaigns ACLOUD, PAMARCMiP, and ARCTAS assuming a fixed $SSA$ of $20\,\mathrm{m^2\,kg^{-1}}$. The separated effects by BC suspended in snow ($\Delta F_\mathrm{snow}$, panel a), atmospheric BC ($\Delta F_\mathrm{atm}$, panel b), and the total effect ($\Delta F_\mathrm{tot}$, panel c) are shown. The daily mean solar radiative effects of BC in cloudy conditions is displayed by shades bars. The difference of $\Delta F_\mathrm{tot}$ between simulations with and without clouds is given in panel d.

cloudy and cloudless conditions are observed, but with different direction. For the ACLOUD case, the clouds reduce the warming effect of BC particles mainly due to a reduction of radiation that reaches the surface. As almost no atmospheric BC is present, only $\Delta F_\mathrm{snow}$ is affected.

For the ARCTAS cases, the clouds always increase $\Delta F_\mathrm{tot}$. For a BC mass concentration of $5\,\mathrm{ng\,g^{-1}}$ even the sign shifts
5 from a total cooling to a total warming effect of BC. For ARCTAS, with high atmospheric BC concentrations, the presence of clouds mainly reduce the cooling effect of the atmospheric BC, $\Delta F_\mathrm{atm}$. As the atmospheric BC layer is located mostly



above the cloud, the radiative effect of the clouds, which is typically much stronger than the absorption by the atmospheric BC, reduces the significance of the atmospheric BC effect. For higher BC mass fractions in the snow, the increase of $\Delta F_{\text{tot}}$ by adding a cloud becomes weaker because $\Delta F_{\text{snow}}$ simultaneously slightly decreases in cloudy conditions.

The PAMARCMiP case, characterized by the low sun elevation, in general, shows a reduced effect by clouds. Here, the reduction of the cooling effect of by atmospheric BC, $\Delta F_{\text{atm}}$, and the increase of the BC snow effect, $\Delta F_{\text{snow}}$, compete each other and result in different total cloud radiative effects. Model runs with and without the upper ice cloud layer did not show any significant difference in $\Delta F_{\text{tot/atm/snow}}$, which allows concluding that mainly the presence of the low liquid water clouds affects the radiative effects of BC particles.

 In summary, the comparison of the radiative effects by BC particles in snow and atmosphere with typical concentrations and

mass fractions observed in Arctic spring and summer are rather small compared to other parameters (SZA, grain size) which are contributing to solar cooling or heating on the surface level. The highest radiative cooling of BC particles is in the range of $1\,\text{W}\,\text{m}^{-2}$ and was estimated for low SZA, high BC particle mass fractions, and large grains.

### 3.2 Vertical radiative impact of BC particles in the atmosphere and snow

#### 3.2.1 Heating rate profiles in the atmosphere

In the atmosphere, BC particles can absorb solar radiation and lead to a local warming effect which might influence the stability of the atmosphere (Wendisch et al., 2008). To quantify these effects for Arctic atmospheric BC particles, profiles of the heating rates were simulated for the three cases ACLOUD, ARCTAS, and PAMARCMiP. Based on simulations with and without atmospheric BC, the total heating rate $HR_{\text{tot}}(z)$ and the effective heating rate of BC particles $HR_{\text{BC}}(z)$ was calculated (see Eqs. 3 and 4) were applied. Figure 7 shows daily averaged profiles of $HR_{\text{tot}}(z)$ and $HR_{\text{BC}}(z)$ calculated for the three BC

profiles. Solid lines represent the cloudless scenarios while dotted lines show simulations where the two predefined cloud layers were added. The location of the clouds is indicated by the gray shaded area. Highest total heating rates in cloudless conditions were found for the ACLOUD case, with maximum values of more than $1.2\,\text{K}\,\text{day}^{-1}$ in about 2-4 km altitude. This altitude range was characterized by enhanced humidity leading to a stronger absorption of solar radiation by the water vapour. The spring campaigns ARCTAS and PAMARCMiP were characterized by lower water vapour concentrations (factor of four

and ten lower than for ACLOUD, respectively) and reduced incident solar radiation due to the time of year and latitude of the observations. This leads to significant lower values of $HR_{\text{tot}}(z)$ compared to the ACLOUD case. While ARCTAS shows a similar vertical pattern with maximum $HR_{\text{tot}}(z)$ of $0.5\,\text{K}\,\text{day}^{-1}$ in the lower troposphere below 5 km altitude, the conditions during PAMARCMiP lead to a maximum of $HR_{\text{tot}}(z)$ of about $0.25\,\text{K}\,\text{day}^{-1}$ located in 5-6 km altitude. This corresponds to the rather dry lower troposphere observed in spring time in the central Arctic. By adding clouds in the simulations, the highest

$HR_{\text{tot}}(z)$ are observed within the liquid water cloud layer, where solar radiation is absorbed by the cloud particles. Similar to the cloudless scenarios, the ACLOUD case shows the highest values of $HR_{\text{tot}}(z)$ with up to $4.1\,\text{K}\,\text{day}^{-1}$) at cloud top of the lower liquid cloud layer. Absorption in the higher ice cloud less pronounces and the increase of $HR_{\text{tot}}(z)$ is significant lower.





The profiles of the effective BC heating rate $HR_{\mathrm{BC}}(z)$ (Fig. 7b) shows a completely different pattern compared to $HR_{\mathrm{tot}}(z)$. In general, $HR_{\mathrm{BC}}(z)$ is about one magnitude lower than $HR_{\mathrm{tot}}(z)$ for all three cases. Significant BC heating rates are observed only for the ARCTAS and PAMARCMiP cases with values up to $0.1\,\mathrm{K\,day^{-1}}$. The profiles of $HR_{\mathrm{BC}}(z)$ are strongly correlated with the vertical distribution of BC particles in the measured profiles. Maximum $HR_{\mathrm{BC}}(z)$ are located in the pollution layers.

The pollution layer observed during PAMARCMiP at 5 km and the BC layers of ARCTAS above 5 km altitude show the largest relative impact of BC particles where nearly one-fifth and one-third, respectively, of the total solar heating is attributed to BC absorption. In lower altitudes of the ARCTAS case, the enhanced absorption by water vapor reduces the relative importance of BC particles. For the summer case of ACLOUD, $HR_{\mathrm{BC}}(z)$ is rather small in all altitudes and does contribute to the total radiative heating by only 10 %. However, in low altitudes, the absolute values of $HR_{\mathrm{BC}}(z)$ are in the same order for both,

ACLOUD and the PAMARCMiP. This illustrates that the effect of a higher BC particle concentration during PAMARCMiP is compensated by the dependence of $HR_{\mathrm{BC}}(z)$ on the amount of the available incoming solar radiation and the atmospheric water vapour concentration.

Adding clouds in the simulations, affects $HR_{\mathrm{BC}}(z)$ of the three cases differently. While the clean atmosphere layer of ACLOUD and the PAMARCMiP cases show almost no differences to cloudless conditions, a minor cloud effect is observed

for the ARCTAS case and the polluted layer of PAMARCMiP. In both cases, the ice cloud leads to a slightly increase of $HR_{\mathrm{BC}}(z)$ by about 5 % within and above the cloud layer. This is caused by the enhanced reflection of the incoming radiation which leads to additional absorption of the reflected radiation by the atmospheric BC particles. In altitudes between the ice and liquid water clouds no significant effect by the clouds are observed. Within and below the liquid water cloud $HR_{\mathrm{BC}}(z)$ is significantly reduced by almost $0.01\,\mathrm{K\,day^{-1}}$ for the ARCTAS case. This cloud effect is caused by the strong reflection of

radiation at the cloud top leading to a reduction of radiation reaching into and below the cloud layer.

Comparing all simulations, it can be concluded that the absolute radiative effects of atmospheric BC particles are potentially strongest in early spring when incoming solar radiation starts to increase and BC particle concentration is still high enough. Furthermore, the surface conditions in spring are dominated by snow and ice coverage which causes an increase in the amount of upward radiation contributing to the atmospheric heating rate. In late spring and summer, the BC particle concentration

decreases rapidly, while the absorption by water vapour becomes more and more dominant with increasing temperatures.

### 3.2.2 Heating rate profiles in the snow pack

To access, in which layers of the snow pack the strongest absorption of solar radiation and, therefore, a potential enhancement of the snow metamorphism is located, profiles of the heating rates within the snow pack $HR_{\mathrm{tot}}(z)$ were calculated. To quantify, how BC particles deposited in snow may change these heating profiles, the effective BC heating rates $HR_{\mathrm{BC}}(z)$ were calculated

for different single layer and multi-layer scenarios as introduced in Tab. 3. For the atmospheric boundary conditions, the ACLOUD case was chosen. Figure 8 shows $HR_{\mathrm{tot}}(z)$ (panel a) and $HR_{\mathrm{BC}}(z)$ (panel b) for all cases in the first 20 cm of the snow pack. Below, the transmittance is less than 1 % and all heating rates become negligible, which is in agreement with Järvinen and Leppäranta (2013). For all cases, the total heating rate quickly decreases with depth. The simulation for the single





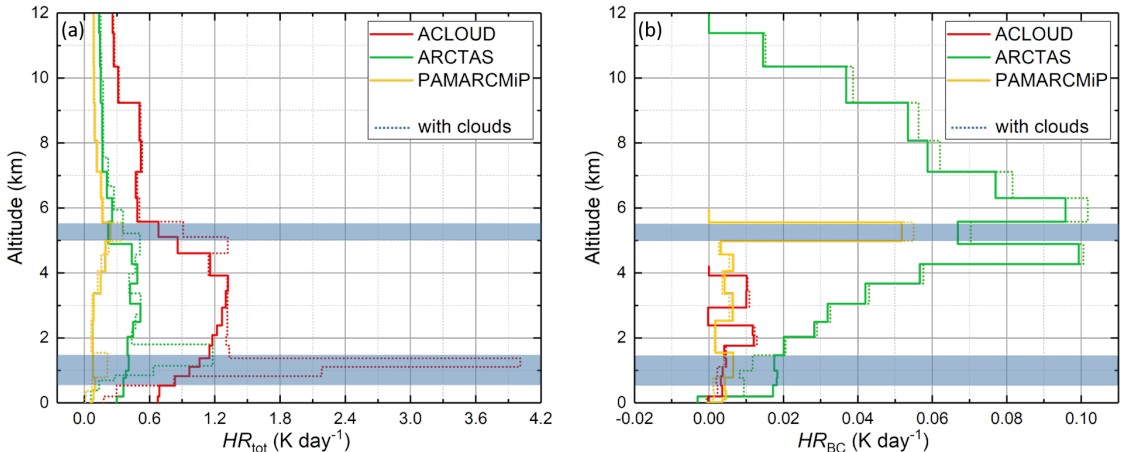

**Figure 7.** Daily averaged profiles of the total radiative heating rate $HR_{tot}(z)$ (panel a) and the effective BC heating rate $HR_{BC}(z)$ (panel b) calculated for the three cases ACLOUD, ARCTAS, and PAMARCMiP. Both, cloudless simulations (solid lines) and cloudy scenarios (dotted lines) are shown. The gray shaded areas indicate the location of the cloud layers.

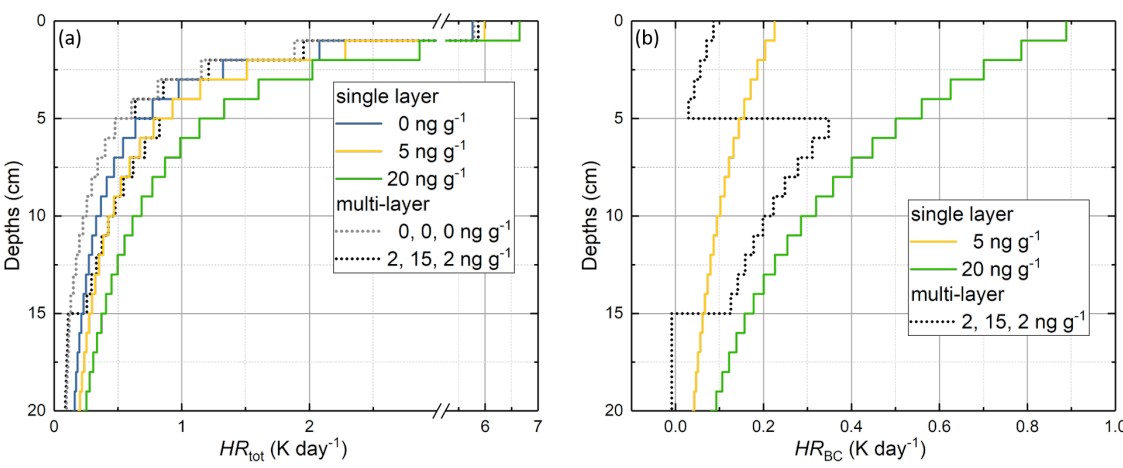

**Figure 8.** (a) Daily mean total heating rate profiles within a snow pack $HR_{tot}(z)$ for three single layers and one multi-layer case assuming ACLOUD conditions. (b) Corresponding effective BC heating rate profiles $HR_{BC}(z)$.

layer (solid lines) snow pack shows the maximum values of $HR_{tot}(z)$ which are located in the top most layers and reach values up to $6.6\,\mathrm{K\,day}^{-1}$ (note, the scale break in Fig. 8a).





Assuming different BC mass fractions in the single layer case, slightly increases $HR_{\text{tot}}(z)$ in the entire column. In the multi-layer case, this increase is limited to the upper part of the profile. This contribution of BC particles to the total radiative heating is quantified by $HR_{\text{BC}}(z)$ and shown in Figure 8b. Largest $HR_{\text{BC}}(z)$ are observed for the most polluted single layer case with a BC mass fraction of $20\,\text{ng}\,\text{g}^{-1}$. For this case, the contribution by BC particles amounts to almost $0.9\,\text{K}\,\text{day}^{-1}$ in the

top most layer dropping down to a value of less than $0.1\,\text{K}\,\text{day}^{-1}$ in $20\,\text{cm}$ snow depth. Compared to the total radiative effect, $HR_{\text{BC}}(z)$ contributes with about $15\,\%$ to the heating rate at the top snow layer and $40\,\%$ to the heating in the base layer. For the typical Arctic BC mass fraction of $5\,\text{ng}\,\text{g}^{-1}$, this contribution of BC particles is significantly lower ranging between $3\,\%$ and $20\,\%$.

The multi-layer cases is characterized by smaller snow grains in the top layer ($SSA = 40\,\text{m}^2\,\text{kg}^{-1}$) compared to the single-

layer cases ($SSA = 20\,\text{m}^2\,\text{kg}^{-1}$) and, therefore, shows reduced values of $HR_{\text{tot}}(z)$. According to the structure of the snow pack, $HR_{\text{BC}}(z)$ is largest in the layer of the highest BC mass fraction. Beneath this layer ($z < 15\,\text{cm}$) the heating rates for the pristine and polluted case are almost similar ($HR_{\text{BC}}(z) = 0\,\text{K}\,\text{day}^{-1}$). In this base layer, the largest snow grains are assumed (lowest $SSA$) which increases the absorption of radiation by the snow ice water.

Based on these results, it becomes evident that the absorption of solar radiation by the ice water of the snow grains dominates

the total heating rate in the snow pack, especially at the top layer, where most radiation is absorbed. Therefore, the snow grain size typically plays a larger role than the concentration of BC particles suspended in snow. This illustrates, that the snow metamorphism is a self amplifying effect and can only slightly be accelerated by BC particles.

Simulations in cloudy conditions (not shown here), result in a reduced $HR_{\text{tot}}(z)$ and $HR_{\text{BC}}(z)$ because the clouds reduce the incoming solar radiation. Similarly, a change of the solar zenith angle affects the results by changing the available solar

radiation. Therefore, the ACLOUD case used in the simulations presented in this section represents the maximum radiative effects compared to ARCTAS and PAMARCMiP conditions. In general, it can be concluded that the solar heating by BC particles suspended in the snow pack is most effective for low SZA (spring and summer conditions with high amount of available incoming radiation), decreasing $SSA$ (aged snow in conditions near melting temperature), and increasing BC particle mass fractions (accumulated BC particles caused by melting). Such conditions are mostly linked to late spring and summer,

when the Sun is high, snow is close to melting and BC has accumulated. This suggests that the maximum heating rates due to atmospheric BC and BC suspended in the snow pack typically occur in different periods of the year, early spring and early summer, respectively.

## 4   Summary and conclusions

This study analyzed the direct solar radiative effect of Arctic BC particles (suspended in the atmosphere and in the snow pack)

under cloudless and cloudy conditions. For this purpose, an atmospheric and a snow radiative transfer model were coupled to account for the radiative interactions between both compartments. Typical atmospheric BC vertical profiles and BC particle mass fractions in the snow pack, derived from three field campaigns in the American and Atlantic Arctic, ACLOUD, ARCTAS, and PAMARCMiP, were used to set up the simulations. The BC radiative effects were quantified by the surface radiative effect





and profiles of heating rates in the atmosphere and the snow, which were presented on the basis of daily averages. For the surface radiative effect, the contribution by atmospheric and snow BC particles was separated. For the heating rate profiles, the effective contribution of BC particles to the total heating rates was derived and compared to other parameters leading to a warming or cooling (e.g., water vapour, clouds, snow grain size).

The simulations suggest, that the local radiative effects of BC are small compared to the radiative impact of these other parameters. Therefore, the significance of the BC radiative effects show a strong seasonal dependence. In cloudless conditions, the atmospheric water vapour is a much stronger driver of the atmospheric heating rates as compared to BC particles. In summer (ACLOUD) and in lower latitudes (ARCTAS), the Arctic shows the most humid conditions, where absorption of water vapor dominates over the BC radiative effects. Similarly, the available solar radiation limits the magnitude of the BC radiative effects.

Despite the more polluted atmosphere, the low solar zenith angle of the cases of PAMARCMiP (high latitude) and ARCTAS (spring season) did show lower BC radiative effects than the ACLOUD case. Thus, in the Arctic, the BC radiative effect is about a magnitude lower than observed in lower and tropical latitudes, where also the pollution level is typically higher. For example, studies conducted for northern India or China reported on BC heating rates in the atmosphere larger than $2\,\mathrm{K\,day^{-1}}$ (Tripathi et al., 2007; Wendisch et al., 2008). Compared to the maximum heating rate of $0.1\,\mathrm{K\,day^{-1}}$ derived in this study

for Arctic conditions, such high values can influence the lapse rate feedback and the atmospheric stability. However, for the rather pristine Arctic, the calculated daily mean BC heating rates derived in this study are small, and, thus, BC particles can not significantly modify the atmospheric stability. However, in other Arctic regions characterized by a higher atmospheric BC particle concentrations due to local fires, e.g., northern Siberia, a stronger impact can be expected.

    Similarly, the mass fraction of BC particles suspended in the Arctic snow pack is by magnitudes lower than observed in alpine

snow in lower latitudes. Accordingly, the absorption of radiation by the snow water itself dominates the radiative warming in the snow pack. For typical conditions of the central Arctic, the absorption due to BC particles contributes only with 3 % to the total heating rate in the uppermost snow layer. These results indicate, that the microphysical properties of the snow pack (mainly snow grain sizes) are a more important drivers for the degree/strength of the snow metamorphism.

    These dependencies of the relevance of the BC radiative effects suggests that the maximum heating rates due to atmospheric

BC and BC suspended in the snow pack typically occur in different periods of the year. While atmospheric BC shows the largest effect in early spring (high concentration of atmospheric BC, medium high Sun, low water vapour), the BC particles suspended in snow warm more effective in early summer (accumulation of BC particles in snow, high Sun, large snow grain size).

    Radiative transfer simulations assuming cloudless and cloudy conditions were compared to estimate the role of clouds on

the surface warming/cooling by BC particles and the BC heating rates. Clouds reflect the incoming radiation and, therefore, reduce the available radiation reaching the lower altitudes. This reduces the potential of the warming effect by BC particles suspended in the snow. Similarly, the cooling effect by atmospheric BC on the surface radiative budget is weakened in the presence of clouds. The competition of these both cloud effects depends on the BC concentrations in the snow and atmosphere and is affected by the increased broadband surface albedo and the multiple scattering in presence of a cloud layer. The profiles

of the effective BC heating rate are mainly affected by the ice cloud in higher altitude. Within and above the cloud, the radiation



reflected by the cloud enhances the local radiative heating by BC. Contrarily, a low liquid water cloud reduces the available incoming radiation, such that the effective BC radiative effect is lower for the cloudy case compared to the cloudless case. For the same reason, the presence of clouds reduces the radiative heating rates within the snow pack.

For the Atlantic Arctic, based on the presented study, we therefore conclude that: (i) the warming effect of BC suspended in the snow overcompensated the atmospheric BC cooling effect, (ii) the impact of clouds shows an opposite direction for atmospheric BC cooling and snow BC warming, and (iii) the BC radiative effect is of minor importance compared to other drivers. However, for the expected increase of BC particle mass concentrations in the future, the relative importance of BC particles will need to be reevaluated. Additionally, ongoing research, e.g. by using results expected from the currently ongoing MOSAiC drift experiment and associated flight measurements, will allow to quantify the effective radiative effect of BC also in the Eastern and Central Arctic using the methods proposed here.

*Data availability.* Atmospheric BC mass concentrations for the ARCTAS campaign are available on https://www-air.larc.nasa.gov/cgi-bin/ArcView/arctas, last access: 22 January 2020 (PI: Yutaka Kondo). PAMARCMiP and ACLOUD data are available on PANGAEA (https://doi.pangaea.de/10.1594/PANGAEA.899508 and https://doi.org/10.1594/PANGAEA.899937, respectively)

*Author contributions.* All authors contributed to the editing of the manuscript and to the discussion of the results. MW, AE, and AH designed this study. TD drafted the manuscript, performed the radiative transfer simulations and prepared the figures. AE, EJ, and BH contributed to the interpretation of the radiative transfer simulations. MZ processed the SP2 data. JS compiled the atmospheric BC profiles.

*Competing interests.* The authors declare that they have no conflict of interest.

*Acknowledgements.* We gratefully acknowledge the funding by the Deutsche Forschungsgemeinschaft (DFG, German Research Foundation) - Project-ID 268020496 - TRR 172, within the Transregional Collaborative Research Center "ArctiC Amplification: Climate Relevant Atmospheric and SurfaCe Processes, and Feedback Mechanisms (AC)³". We thank Sho Ohata and Makoto Koike for providing the atmospheric BC profiles measured during the PAMARCMiP campaign 2018.



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
