# Peer review of "Combining atmospheric and snow radiative transfer models to assess the solar radiative effects of black carbon in the Arctic"

_Atmospheric Chemistry and Physics, 2020_

## Referee Comment (RC1) · Quentin Libois (Referee) · 27 Feb 2020

Review of « Combining atmospheric and snow layer radiative transfer models to assess the solar radiative effects of black carbon in the Arctic », by Tobias Donth et al.

**General comments**

This study aims at estimating the radiative impact of black carbon (BC) particles suspended in the atmosphere and contained in the snowpack in the Arctic. It simultaneously and consistently computes the radiative forcing of BC in both the snowpack and the atmosphere. To this end it couples an atmospheric and a snow radiative transfer model. The BC atmospheric concentrations are taken from three aircraft campaigns that explored various atmospheric conditions, from early spring to summer. A variety of radiative transfer simulations are performed, where snow properties and BC mass concentrations are varied to cover the range of Arctic conditions reported in the literature. The main conclusion is that the radiative impact of BC is marginal in typical Arctic conditions, amounting to about a few percent of the total heating rates and to less than 1 W m$^{-2}$ in terms of surface forcing. The authors also point a competition between shading of the surface by atmospheric BC that counteracts the warming effect of BC in snow. The impact of clouds is investigated, also showing complex interactions, where depending on their altitude and optical thickness, clouds can either enhance the effect of BC through multiple scattering, or reduce it by shading. In any case, the authors highlight that other drivers of the Arctic energy budget are more significant than BC, such as absorption by water vapour, snow metamorphism and clouds.

The topic of the study is relevant to *ACP* because it combines numerical simulations and field observations to provide a geophysical analysis. The paper is well written and easy to follow. There is much relevant physical insight and the conclusions are drawn rigorously from the computations. The findings are not a breakthrough but they have the merit to provide a self-standing investigation of the total BC impact in Arctic conditions, where previous studies have either focused on the atmosphere or in the snow. This is probably the greatest added value compared to previous work. We may regret the lack of field data for the snow. Likewise, the fact that only offline radiative computations are performed precludes a rigorous quantification of the impacts on atmospheric dynamics and snow evolution. As a consequence, the numerous conclusions on the impact of BC with a dynamic perspective appear quite weak and should be better motivated with appropriate references. Practically, data from aircraft campaigns are only used to derive average profiles of temperature, humidity (in a manner that should be more detailed) and BC, but snow properties are chosen based on other studies and more as varying parameters. This is not an inappropriate approach but this makes the importance of novel data quite limited in this work. Based on the comments above, I recommend this paper be published after the corrections suggested below are tackled.

**Specific comments**

1) It is clear that the study focuses on BC and the conclusion is that BC is not so critical with the amounts currently observed in the remote Arctic. However, recently there have been plenty of studies clearly showing altered surface albedos because of light absorbing impurities. The latter could then be dust, micro-organisms or anything else. It might be worth insisting that you only deal with BC, which is one amongst many others light absorbing species, so that the conclusion should not be over-interpreted as « there is no impact of impurities in the Arctic ». Likewise, the geopgraphical area to which the work is relevant should be better identified.

2) The paper focuses on energy budgets (of the atmsophere and snow). Although the impact of BC on these budgets is very limited, BC strongly impacts the light penetration depth in snow, or equivalently snow transmittance. For instance, if a 20 cm snow layer in the Arctic has a transmittance of 1 %, adding BC may decrease this value down to 0.5 %. This is nothing for the

snow budget, buth this makes a huge difference for the amount of energy transmitted. This will for instance be critical for photosynthesis within or under the snowpack. Maybe this should be mentioned somewhere so that again readers don't think « BC does not matter ».  The paper by Tuzet et al., (2019) may be a useful reference for that.

3) The iterative coupling between libRadtran and TARTES is a first valuable step towards consistent radiative transfer simulations. I can only encourage the authors to fully incorporate the scattering snowpack in libRadtran for their future work. This can be done simply by providing the single scattering properties computed by TARTES to create new « atmospheric » layers in libRadtran which would be extremely thin. Such strategy would avoid the iterative coupling and be overall more elegant. See for instance Blanchet and List (1987) for a very similar study.

4) The evaluation of BC contribution to heating rates or total absorption is sometimes misleading. The authors often conclude that BC contribution being a few percents its impact is negligible. However think in terms of $CO_2$ forcing, where a few W $m^{-2}$ (in addition to hundreds of W $m^{-2}$) can fully change the face of the Earth. I simply mean that it is hard to conclude that a few percents perturbation of the energy budget due to BC is insignificant. Be more cautious in the conclusions, unless you have strong and better argumented reasons to think that it is indeed negligible.

5) The paper somehow lacks a bit of discussion, where the limits of the study and recommendations for future work would be provided. In particular, the importance of BC in locations where it is much more concentrated could be discussed. The representativity of the BC atmospheric profiles used as well. The use of daily averages to asses a radiative impact may not be relevant (maximum values matter as well). The link with snow metamorphism is only qualitative why models allow to explicitly simulate the impact of these heating profiles of metamorphism, etc. All these points should be brought to the reader and further investigated in future work, if not already further discussed in the present paper.

**Technical corrections**

title : would **forcing** be more appropriate than « effects » ? Consider also removing « layer »

**p.1**

l.1 : the absrtact could be written using the present. More generally there is no conssitent use of present or past in the manuscript. Some homogenization would be recommended.

l.2 : BC particles are not really « suspended » in the snowpack. They're rather embedded or contained. Consider changing this throughout the paper.

l.2 : « by » → using

l.4 : « interactions » is unclear. Maybe use multiple scattering or coupling

l.4 : « a snow layer » should be replaced by « a snow » because multi-layer snowpacks are explored. Maybe write « An atmospheric and a snow radiative... »

l.6 : this radiative effect is very dependent on the SZA chosen. Please clarify

l.9 : counteracting «effect »

l.10 : technically snow density also impacts snow optical properties

l.10 : « however » does not really oppose to anything

l.12 : I think « ice » could be used instead of « ice water »

l.19 : absorbs, scatters

L.24 : predominantly

**p.2**

l.1 in higher → to higher

l.4-5 : using « nevertheless » and « still » in two consecutive sentences makes it diffcult to follow

l.7 : « of suspended » is awkward

l.9 : can be expected → are observed

l.14 : double « the »

l.20 : associated with → , thus increasing the amount…

l.29 : there is no « novel » feedback described here. BC is just shown to trigger the snow metamorphism feedback. There is actually a feedback because BC impact will be stronger for lower SSA, but this should be described here if this is what you actually mean.

l.34 : warming

l.34 : « the atmospheric layer containing BC »

**p.3**

l.4-10 : this paragraph is not very clear and could probably be removed

l.15 : why only « local » ? Not clear whether this refers to local pollution or not

l.16 : With

l.15 : « interactions » is not very appropriate

**p.4**

l.5 : « relevance » is not well chosen → contribution

l.7 : « setup » suggests there is some evolution from an initiation which is not the case. « Configuration » would be better.

l.8 : change title to «BC profiles from aircraft campaigns »

 l.9 : not clear what this « atmospheric » model is

l.23 : are these « snow properties » used later on ?

**p.5**

l.5 : **is** available

l.10 : consider adding some information about the thermodynamical profiles measured during the flights, if actually used further

l.13 : url for libRadtran download should be provided here or in the Data availability section

l.15 : is reference to « Evans 1998 » relevant here ?

l.15 : precise that this assumes a plane-parallel atmosphere

l.19 : mention explicitly humidity (or water vapor)

**p.6**

l.1 : « adjusted » is unclear. Do you mean that profiles from the mid-campaign were used ?

l.7 : where do the cloud optical properties come from ?

l.9 : can you provide optical thickness values ?

**p.7**

l.2 : provide url for TARTES

l.6 : The reference provided is not about delta-Eddington approximation. Prefer Joseph et al. (1976)

l.8 : SSA should not be italic (throughout the text)

l.9 : there are two shape parameters (B and g). Please provide the values used.

l.12 : ot → to

l.13 : another important point is that impurities are assumed to be Rayleigh scatterers

l.23 : please provide some references for the SSA values assumed

l.33 : SSA for fresh snow could be larger

**p.8**

l.5 : no, spectral albedo does not depend on the spectral distribution of irradiance. Broadband albedo does

l.7 : « shifts » suggests a conversion of some wavelengths to some others. Maybe say « filters/absorbs longer wavelengths so that the downward irradiance spectrum is shifted towards shorter wavelengths »

l.9-10 : for which snowpack ?

**p.9**

Figure 2 : « adjusted » parameters is not clear. Do you mean that they can vary ? Maybe specify that the procedure is done at high spectral resolution so that the figure holds for a single wavelength. Consider adding a title with TARTES/libRadtran (or SNOW/ATMOSPHERE) on top of the colored boxes.

l.4 : surface radiative effect is not clear (radiative forcing ?)

l.7 : specify what the default cases are when snowpack is considered (what BC in atmosphere ?) or atmosphere is considered (what BC in snow ?)

l.11 : can ? Should be « does » ?

**p.10**

l.5-7 : the details about the vertical resolution of both radiative transfer codes should be given earlier in the presentation of the models configurations.

l.13 : daily means may hide much larger instantaneous values which are very relevant both for snow metamorphism and atmospheric dynamics. Adding the max values on the subsequent plots would be very useful

**p.11**

l.2 : should be HR_BC ?

l.6 : what kind of dependence ?

l.6 : « respectively » is awkward

l.15 : I don't see any zoom of the Figure 4, but definitely this would be useful

l.31 : please specify that this is the impact of BC **on the broadband albedo**. Other optical quantities might be much more altered

**p.12**

Table4 : particle

l.3 : the distinction between titles 3.1.1 and 3.1.2 is not obvious. Say radiative **forcing** ?

l.5 : twice « effect »

l.5 : **first** calculated… (because standard is with daily cycle). Note also that sometimes the past is used, sometimes the present. It'd be worth homogenizing this.

**p.13**

l.8 : are **then** analyzed

**p.14**

l.1 : how much in %? Using relative contributions rather than absolute forcing may be instructive to compare campaigns. This holds also when evaluating the contribution of clouds. Of course they shade the surface, but how does the relative forcing of BC vary ?

**p.16**

l.5 : of

l.15-16 : already said in the introduction

l.19 : remove « were applied »

l.32 : « is less pronounced … significantly »

**p.17**

l.2 : one order of

l.15 : slight

l.27 : « to access » is awkward

l.32 : « transmittance » is unclear. Do you mean irradiance with respect to surface irradiance ? Then relative illumination or relative irradiance is better.

l.33 : what is « quickly » for a heating rate decrease?

**p.19**

l.2 : I'm surprised not to see the shading of the lower layers by BC in the topmost layers. Did you observe that below ?

l.12 : = 0 or ≈ 0 ?»

l.17 : I think the conversion from a contribution to a snow heating rate into a metamorphism rate is not that straightforward, especailly with daily means. Providing snow physics references would be helpful to strengthen your conclusions

l.33 : were used **in** the

**p.20**

l.3 : « other parameters » is awkward. Please clarify. Maybe mention reference/unpolluted conditions

l.5 : again, « local » is unclear

l.6 : why « therefore » ?

l.6 : show**s**

l.7 : « driver » means that its variability controls the variability of the heating rates. Is that the case (then it should be detailed) ? You could have varying BC for constant water vapour, then the variations of the heating rates would be driven by BC.

l.15 : lapse-rate feedback refers to the response of the atmsophere to a surface temperature change. In terms of vertical gradient of temperature. I'm not sure you really mean this here (as a feedback).

l.16-17 : again, what is « small » ? can you provide elements to support the fact that 0.1 K day$^{-1}$ cannot change atmospheric stability ?

l.17-18 : this could be moved to the introduction, that the study focuses on remote Arctic locations, not on locally polluted areas.

l.33 : these **two** cloud

**p.21**

l.4 : « Atlantic Arctic » should be emphasized in the introduction

l.5 : cooling is at the surface, please clarify

l.10 : some elements should be provided about other types of impurities which may eventually be more critical than BC in the Arctic

**References**

BLANCHET, J. P., & List, R. (1987). On radiative effects of anthropogenic aerosol components in Arctic haze and snow. *Tellus B*, *39*(3), 293-317.

Joseph, J. H., Wiscombe, W. J., & Weinman, J. A. (1976). The delta-Eddington approximation for radiative flux transfer. *Journal of the Atmospheric Sciences*, *33*(12), 2452-2459.

Tuzet, F., Dumont, M., Arnaud, L., Voisin, D., Lamare, M., Larue, F., ... & Picard, G. (2019). Influence of light-absorbing particles on snow spectral irradiance profiles. *Cryosphere*, *13*(8).

---

## Referee Comment (RC2) · Anonymous Referee #2 · 26 Apr 2020

**Review of the manuscript 'Combining atmospheric and snow layer radiative transfer models to assess the solar radiative effects of black carbon in the Arctic' by Donth et al.**

The manuscript describes the use of BC measurements in the atmosphere and the snow, to calculate the radiative effect of BC applying an iterative procedure to couple the radiative transfer in the atmosphere and snow.

The manuscript is well-organised and include detailed description of the results. It has, however, some points to be clarified before publication.

**Comments**

- **Page 3, caption table 1**: There are various ways of defining BC. Please include a reference to for example Petzold et al. (2013) to clearly define your use of BC. Also please mention how EC values compare with BC values.

- **Page 4, lines 17-19**: Sentence is unclear. Please reformulate.

- **Page 5, line 15**: The reference to Evans (1998) and the SHDOM code appears to be out of place. Should it be Stamnes et al. (1988) instead?

- **Page 5, lines 12-18**:

  1. How many streams was used for DISORT?
  2. The solar zenith angle is large for all regions considered. Did you make any spherical corrections? If not, why not, and how do you expect this to affect your results?
  3. What was the vertical resolution of your model atmosphere?

- **Page 6, Fig 1**: The profiles shown are averages. Please also include the standard deviation (or other measure of variability) of the profiles to give an idea of how the profiles varied for the different campaigns.

- **Page 7, line 2**: In the snow a two-stream model is used. Presumably more streams were used for the atmospheric radiative transfer. Why is it sufficient to use only two streams in the snow pack?

- **Page 7, line 6**: Stamnes et al. (1988) is not a reference for the delta-Eddington approximation. Maybe rather cite Joseph et al. (1976)?

- **Page 7, line 14**: In the snow the BC optical properties are from Bond et al. (2013) while in the atmosphere they are from Hess et al. (1998). Hence, the BC particles are different in the atmosphere and the snow. What is the rationale behind this choice other than what is available in the models used?

- **Page 8, Table 3**: Should the first row in the table be named "Thickness" instead of "Depth"?

- **Page 8, lines 20-21**: The sentence "This procedure is repeated until the deviation between previous (step n) and revised surface albedo decrease below 1 %" is unclear. Please reformulate.

- **Page 10, Fig. 3**: May it be concluded from the plot that the iteration procedure as no impact on the surface albedo in the wavelength region where BC absorbs?

- **Page 11, lines 1-2**: The upward and downward irradiances were averaged and from these the averaged heating rates were calculated. This appears as a rather unusual and unphysical approach. Would it not be more appropriate to calculate the instantaneous heating rates and then average these?

- **Page 11, line 15**: The enlargement of Fig 4. seems to be missing. As Fig. 4 is, it does not make sense to have many overlapping lines. Please provide a zoom in of the visible wavelength region ($\lambda < 700$ nm).

- **Page 11-12, line 31-1**: In the introduction it is stated that "the radiative effects of atmospheric BC particles and BC suspended in snow shows an opposite behavior" and "these two effects balance each other". Here it says "the impact of BC particles suspended in the snow pack is assumed to be of minor importance for Arctic conditions". These statements appears to be contradicting each other. Please clarify.

- **Page 12, lines 1-2**: The paper by Warren (2013) discussed remote sensing of BC in the snowpack. I can not see how it justifies the claims made here?

- **Page 14, line 2**: Is the factor of about 3 mostly due to differences in solar zenith angle?

- **Page 16, line 32**: Sentence starting with "Absorption in the ..." is unclear. Please reformulate.

- **Pages 19-21**: In the conclusions please discuss how the results from this study compare with previous studies mentioned in the introduction.

**Language corrections**

- **Page 2, line 7**: change 'of suspended' to 'suspended'.

- **Page 2, line 34**: change 'will warming' to 'will warm'.

- **Page 3, line 16**: remove '.' after 'quantified.'.

- **Page 9, line 1**: Should it be "converge" instead of "conversion"?

- **Page 11, line 6**: Remove "(SSA respectively)".

**References**

T. C. Bond, S. J. Doherty, D. W. Fahey, P. M. Forster, T. Berntsen, B. J. DeAngelo, M. G. Flanner, S. Ghan, B. Krcher, D. Koch, S. Kinne, Y. Kondo, P. K. Quinn, M. C. Sarofim, M. G. Schultz, M. Schulz, C. Venkataraman, H. Zhang, S. Zhang, N. Bellouin, S. K. Guttikunda, P. K. Hopke, M. Z. Jacobson, J. W. Kaiser, Z. Klimont, U. Lohmann, J. P. Schwarz, D. Shindell, T. Storelvmo, S. G. Warren, and C. S. Zender. Bounding the role of black carbon in the climate system: A scientific assessment. *Journal of Geophysical Research: Atmospheres*, 118(11):5380–5552, 2013. ISSN 2169-8996. doi: 10.1002/jgrd.50171. URL http://dx.doi.org/10.1002/jgrd.50171.

M. Hess, P. Koepke, and I. Schult. Optical properties of aerosols and clouds: The software package OPAC. *Bulletin of the American Meteorological Society*, 79:831–844, 1998.

J. H. Joseph, W. J. Wiscombe, and J. A. Weinman. The Delta–Eddington approximation for radiative flux transfer . *J. Atmos. Sci.*, 33:2452–2459, 1976.

A. Petzold, J. A. Ogren, M. Fiebig, P. Laj, S.-M. Li, U. Baltensperger, T. Holzer-Popp, S. Kinne, G. Pappalardo, N. Sugimoto, C. Wehrli, A. Wiedensohler, and X.-Y. Zhang. Recommendations for reporting "black carbon" measurements. *Atmospheric Chemistry and Physics*, 13(16):8365–8379, 2013. doi: 10.5194/acp-13-8365-2013. URL `http://www.atmos-chem-phys.net/13/8365/2013/`.

Knut Stamnes, Si-Chee Tsay, Warren Wiscombe, and Kolf Jayaweera. Numerically stable algorithm for discrete–ordinate–method radiative transfer in multiple scattering and emitting layered media. *Appl. Opt.*, 27:2502–2509, 1988.

Stephen G. Warren. Can black carbon in snow be detected by remote sensing? *Journal of Geophysical Research: Atmospheres*, 118(2):779–786, 2013. ISSN 2169-8996. doi: 10.1029/2012JD018476. URL `http://dx.doi.org/10.1029/2012JD018476`.

---

## Author Comment (AC1) · 15 Jun 2020

**Reply to Quentin Libois (Reviewer #1):**

**We gratefully thank the reviewer for the detailed review and the numerous larger and smaller suggestions. The comment guided us easily to improve the manuscript. We would like to highlight the efforts of the reviewer, for reading the manuscript very carefully and identifying many typos.**

**Detailed replies on the reviewer's comments are given below. Our replies are given written with indention. Citations from the revised manuscript are given in italic and quotation marks.**

**General comments**

This study aims at estimating the radiative impact of black carbon (BC) particles suspended in the atmosphere and contained in the snowpack in the Arctic. It simultaneously and consistently computes the radiative forcing of BC in both the snowpack and the atmosphere. To this end it couples an atmospheric and a snow radiative transfer model. The BC atmospheric concentrations are taken from three aircraft campaigns that explored various atmospheric conditions, from early spring to summer. A variety of radiative transfer simulations are performed, where snow properties and BC mass concentrations are varied to cover the range of Arctic conditions reported in the literature. The main conclusion is that the radiative impact of BC is marginal in typical Arctic conditions, amounting to about a few percent of the total heating rates and to less than 1 W m-2 in terms of surface forcing. The authors also point a competition between shading of the surface by atmospheric BC that counteracts the warming effect of BC in snow. The impact of clouds is investigated, also showing complex interactions, where depending on their altitude and optical thickness, clouds can either enhance the effect of BC through multiple scattering, or reduce it by shading. In any case, the authors highlight that other drivers of the Arctic energy budget are more significant than BC, such as absorption by water vapour, snow metamorphism and clouds.

The topic of the study is relevant to ACP because it combines numerical simulations and field observations to provide a geophysical analysis. The paper is well written and easy to follow. There is much relevant physical insight and the conclusions are drawn rigorously from the computations. The findings are not a breakthrough but they have the merit to provide a self-standing investigation of the total BC impact in Arctic conditions, where previous studies have either focused on the atmosphere or in the snow. This is probably the greatest added value compared to previous work. We may regret the lack of field data for the snow. Likewise, the fact that only offline radiative computations are performed precludes a rigorous quantification of the impacts on atmospheric dynamics and snow evolution. As a consequence, the numerous conclusions on the impact of BC with a dynamic perspective appear quite weak and should be better motivated with appropriate references. Practically, data from aircraft campaigns are only used to derive average profiles of temperature, humidity (in a manner that should be more detailed) and BC, but snow properties are chosen based on other studies and more as varying parameters. This is not an inappropriate approach but this makes the importance of novel data quite limited in this work. Based on the comments above, I recommend this paper be published after the corrections suggested below are tackled.

> Again, we thank the reviewer for summarizing the open issues of the original manuscript. The replies on the following specific comments hopefully consider also the general concerns raised by the reviewer.

**Specific comments**

1) It is clear that the study focuses on BC and the conclusion is that BC is not so critical with the amounts currently observed in the remote Arctic. However, recently there have been plenty of studies clearly showing altered surface albedos because of light absorbing impurities. The latter could then be dust, micro-organisms or anything else. It might be worth insisting that you only deal with BC, which is one amongst many others light absorbing species, so that the conclusion should not be over-interpreted as ≪ there is no impact of impurities in the Arctic ≫. Likewise, the geopgraphical area to which the work is relevant should be better identified.

> We agree with the reviewer, that the estimates of the BC radiative effect calculated in our study cannot be generalized. Other impurities might give a more significant signal. Also we restricted our analyses to snow on sea ice. The Effects of BC might accumulate as BC particles accumulate when snow melts and bare sea ice is left. Ever more important is BC on glaciers where the accumulation does last more than the 1-3 years before sea ice typically melts. In the revised manuscript we emphasized the limitations of our calculations at several instances:
>
> *"For the conditions over the Arctic Ocean analyzed in the simulations, it is found, that…"*
>
> *"This study analyzed the instantaneous solar radiative effect at the surface of Arctic BC particles (suspended in the atmosphere and embedded in the snow pack) over the sea ice covered Arctic Ocean."*
>
> *"It needs to be considered, that this picture might change if the accumulation of BC particles is more efficient than it is over the snow covered Arctic sea ice, where the sea ice and snow pack does not last more than one to three years. Accumulation of BC on e.g. the Greenlandic glaciers will amplify the radiative forcing on a local scale. Furthermore, BC particles are not the only light absorbing impurities, which are transported into the Arctic. The relevance of dust particles and micro-organisms is currently subject of the scientific discussion and may exceed the effect of BC particles (Kylling et al., 2018, Skiles et al., 2018)."*

2) The paper focuses on energy budgets (of the atmosphere and snow). Although the impact of BC on these budgets is very limited, BC strongly impacts the light penetration depth in snow, or equivalently snow transmittance. For instance, if a 20 cm snow layer in the Arctic has a transmittance of 1 %, adding BC may decrease this value down to 0.5 %. This is nothing for the snow budget, buth this makes a huge difference for the amount of energy transmitted. This will for instance be critical for photosynthesis within or under the snowpack. Maybe this should be mentioned somewhere so that again readers don't think ≪ BC does not matter ≫. The paper by Tuzet et al., (2019) may be a useful reference for that.

> Thanks for pointing at this relevant aspect which we did not consider so far. Indeed, our simulations show a significant decrease of transmissivity below the snow layer. For the homogeneous snow layer, the transmissivity in 20 cm depth for the unpolluted case is about 0.3, while adding a BC concentration of 5 ng $g^{-1}$ reduces the transmissivity to almost 0.2. This obviously may have an impact on the radiative processes in and below the sea ice. In the revised manuscript, we added an additional panel to Figure 8 showing the transmissivity profile and added a short discussion.

[Figure]

Fig. 8a: Transmissivity profiles of solar radiation within the snow pack for three single layers and one multi-layer case assuming ACLOUD conditions.

"Figure 8a shows the transmissivity profiles of solar radiation within the snow pack. The homogeneous single layer reference case without BC particles (SSA = 20 m$^2$ kg$^{-1}$) illustrates the general decrease of transmissivity, which is reduced to 0.3 in 20 cm snow depth. Adding a typical Arctic BC concentration of 5 ng g$^{-1}$ reduces the transmissivity to almost 0.2. This obviously may have an impact on the radiative processes below the snow pack, in and below the sea ice as discussed by, e.g., Tuzet et al (2019) and Marks and King (2014). The inhomogeneous multi-layer case shows in general lower transmissivities due to the enhanced reflection of the smaller snow grains at top of the layer (SSA = 60 m$^2$ kg$^{-1}$ down to 5 cm depth) but also indicates a significant dimming effect of the BC particles."

3) The iterative coupling between libRadtran and TARTES is a first valuable step towards consistent radiative transfer simulations. I can only encourage the authors to fully incorporate the scattering snowpack in libRadtran for their future work. This can be done simply by providing the single scattering properties computed by TARTES to create new ≪ atmospheric ≫ layers in libRadtran which would be extremely thin. Such strategy would avoid the iterative coupling and be overall more elegant. See for instance Blanchet and List (1987) for a very similar study.

We agree, a full coupling of both models is the final goal for further studies combining atmosphere and snow radiative transfer. For this study we first aimed to test, if the coupling is possible in general and how large the interaction is. As the iterative coupling shows a very quick convergence, we concluded that this iterative coupling is sufficient for this study. However, for future studies, we will consider the advice of the reviewer.

4) The evaluation of BC contribution to heating rates or total absorption is sometimes misleading. The authors often conclude that BC contribution being a few percents its impact is negligible. However think in terms of CO2 forcing, where a few W m-2 (in addition to hundreds of W m-2) can fully change the face of the Earth. I simply mean that it is hard to conclude that

a few percents perturbation of the energy budget due to BC is insignificant. Be more cautious in the conclusions, unless you have strong and better argumented reasons to think that it is indeed negligible.

> We agree that also only a few W m$^{-2}$ radiative effect can be significant in terms of the total Arctic wide energy budget. As our study is based on three campaigns, the Arctic wide absolute BC radiative effect (or even forcing) cannot be assed. That's why comparing the W m$^{-2}$ of our study to the $CO_2$ forcing would be misleading. Also because the BC radiative effect is a local instantaneous radiative effect, while the climate effect would include all relevant feedbacks. Therefore, we always compared the BC radiative effect to other radiative effects of other properties, e.g., atmospheric water vapor, clouds, snow grain size. At no point in the manuscript we claim, that the BC effect is negligible for the total energy budget. Our conclusion is, that compared to BC radiative effects, other drivers are more important and these other parameters first need to be constrained more precisely to improve e.g., Arctic climate models.

> In the revised manuscript, we tried to check all our conclusions and adjusted the wording if needed:

> *"The magnitude of solar radiative effects (cooling or warming) of black carbon (BC) particles embedded in the Arctic atmosphere and surface snow layer were quantified on the basis of case studies."*

> *"However, in other Arctic regions characterized by higher atmospheric BC particle concentrations due to local fires, e.g., northern Siberia, a stronger impact can be expected. "*

> *"These results indicate, that the microphysical properties of the snow pack (mainly snow grain sizes) are more important drivers for the degree/strength of the snow metamorphism. It needs to be considered, that this picture might change if the accumulation of BC particles is more efficient..."*

5) The paper somehow lacks a bit of discussion, where the limits of the study and recommendations for future work would be provided. In particular, the importance of BC in locations where it is much more concentrated could be discussed. The representativity of the BC atmospheric profiles used as well. The use of daily averages to asses a radiative impact may not be relevant (maximum values matter as well). The link with snow metamorphism is only qualitative why models allow to explicitly simulate the impact of these heating profiles of metamorphism, etc. All these points should be brought to the reader and further investigated in future work, if not already further discussed in the present paper.

> We agree, that the discussion of our results was lacking in detail. In the revised manuscript we tried to address all issues raised here by the reviewer. As all of the single issues are listed in the technical correction, we did not explicitly reply here and refer to the replies given below.

**Technical corrections**

title : would **forcing** be more appropriate than ≪ effects ≫ ? Consider also removing ≪layer≫

> For the title, we do not think that forcing is appropriate. We calculate the radiative effects (forcing) on the surface radiation budget but also the effect on heating rate profiles. To our understanding, the term "forcing" is linked to the energy budget only. So we would keep "radiative effect" in the title.

> "layer" is removed.

**p.1**

l.1 : the abstract could be written using the present. More generally there is no consistent use of present or past in the manuscript. Some homogenization would be recommended.

> We are sorry that we often struggle with the use of the correct tense. We tried to follow our experience publishing in Copernicus journals, where Copy-Editing mostly changes tense into past, when things are done in past. We tried again to homogenize the text and will hope for advice from the final copy-editing process.

l.2 : BC particles are not really ≪ suspended ≫ in the snowpack. They're rather embedded or contained. Consider changing this throughout the paper.

> Changed to embedded throughout the paper

l.2 : ≪ by ≫ → using

> Changed as suggested.

l.4 : ≪ interactions ≫ is unclear. Maybe use multiple scattering or coupling

> We kept "interactions" as multiple scattering is only one process which is considered when coupling the two models. E.g. also the change of direct to diffuse incoming radiation, which is not driven by multiple scattering alone, changes the radiative properties of the surface.

l.4 : ≪ a snow layer ≫ should be replaced by ≪ a snow ≫ because multi-layer snowpacks are explored. Maybe write ≪ An atmospheric and a snow radiative... ≫

> Changed as suggested.

l.6 : this radiative effect is very dependent on the SZA chosen. Please clarify

> We calculated daily mean values considering the diurnal change of the solar zenith angle. Sure, still the results depend on the location, time of year. We therefore added the minimum solar zenith angle and pointed out that the numbers give daily mean estimates of the BC radiative effects.
>
> *"For pristine early summer conditions (no atmospheric BC, minimum solar zenith angles of 55°) and a representative BC particle mass concentration of 5 ng g$^{-1}$ in the surface snow layer, a positive daily mean solar radiative effect of +0.2 W m$^{-2}$ was calculated for the surface radiative budget."*

l.9 : counteracting ≪effect ≫

> Changed to:
>
> *"The total net surface radiative forcing combining the effects of BC embedded in the atmosphere and in the snow layer strongly depends on the snow optical properties (snow specific surface area and snow density)."*

l.10 : technically snow density also impacts snow optical properties

> Density was added as suggested.

l.10 : ≪ however ≫ does not really oppose to anything

Changed as suggested.

l.12 : I think ≪ ice ≫ could be used instead of ≪ ice water ≫

Changed as suggested.

l.19 : absorbs, scatters

Thanks! changed as suggested.

L.24 : predominantly

Thanks! changed as suggested.

**p.2**

l.1 in higher → to higher

Changed as suggested.

l.4-5 : using ≪ nevertheless ≫ and ≪ still ≫ in two consecutive sentences makes it difficult to follow

We changed the sentences to:

*"In future, a strong intensification of the ship traffic in the Arctic Ocean and further polluting human activities are expected (Corbett et al.,2010). Still, the direct …"*

l.7 : ≪ of suspended ≫ is awkward

We corrected this typo.

l.9 : can be expected → are observed

Changed as suggested.

l.14 : double ≪ the ≫

We corrected this typo.

l.20 : associated with → , thus increasing the amount…

We changed the sentence to:

*"The absorption effect can add to the warming of the atmosphere or the snow pack, when the BC particles are suspended either in the air or embedded in the snow. Furthermore, the BC particles may lead to a reduction of the snow surface albedo if the BC sediments on or into the snow pack (Sand et al. 2013)."*

l.29 : there is no ≪ novel ≫ feedback described here. BC is just shown to trigger the snow metamorphism feedback. There is actually a feedback because BC impact will be stronger for lower SSA, but this should be described here if this is what you actually mean.

Yes, we did not precisely distinguish booth feedback mechanisms. In the revised manuscript, we changed this into:

*"As a further consequence, the absorption by BC particles supports the melting of snow and increases the snow grain size due to an enhanced snow metamorphism, leading to further reduction of the surface albedo. The increase of the snow grain size also feeds back to the absorption by BC particles, which is more efficient for larger snow grain sizes (Warren and Wiscombe, 1980)."*

l.34 : warming

Changed as suggested.

l.34 : ≪ the atmospheric layer containing BC ≫

Changed as suggested.

**p.3**

l.4-10 : this paragraph is not very clear and could probably be removed

As we would like to keep this model aspect in the introduction, we did rewrite the paragraphs as follows to make the statements more clear.

*Several regional and global climate models account for the opposite radiative effects of atmospheric BC particles and snow-embedded BC particles (Samset et al., 2014). However, estimates of the total net forcing rely on the accuracy of the distribution of the BC particles assumed in the particular model. Samset et al. (2014) compared 13 aerosol models from the AeroCom Phase II; all of them included BC. They found that modeled atmospheric BC concentrations often show a spread over more than one order of magnitude. In remote regions, dominated by long range transport, these models tend to overestimate the atmospheric BC particle mass concentrations compared to airborne observations. On the other hand, an underestimation of deposition rates induces a lower BC mass fraction in snow (Namazi et al., 2015). While this may introduce significant local and temporal uncertainties of the BC concentration and related radiative effects, long-term trends and mean multi-model results are representative for Arctic-wide observations (Sand et al., 2017)."*

l.15 : why only ≪ local ≫ ? Not clear whether this refers to local pollution or not

Our aim was to clarify, that our estimates are not general for the entire Arctic. As local obviously can be misleading, we changes the sentence into:

*"On the basis of measured Arctic BC particle mass concentrations for spring and summer months, the instantaneous radiative forcing of BC particles embedded in the snow surface layer and in the atmosphere were quantified for specific cases."*

l.16 : With

Changed as suggested.

l.15 : ≪ interactions ≫ is not very appropriate

We changed this sentence into:

*"With help of the coupled model, the interaction of radiative effects in the atmosphere and the snow pack was considered."*

**p.4**

l.5 : ≪ relevance ≫ is not well chosen → contribution

We changed the last two sentences into:

*"Vertical profiles of heating rates in the atmospheric and in the snow pack are presented for clean and polluted conditions. To estimate the impact of BC particles, effective heating rates are calculated by separating the BC radiative effect from the total heating rates."*

l.7 : ≪ setup ≫ suggests there is some evolution from an initiation which is not the case. ≪ Configuration ≫ would be better.

Changed as suggested.

l.8 : change title to ≪BC profiles from aircraft campaigns ≫

Changed as suggested.

l.9 : not clear what this ≪ atmospheric ≫ model is

We changed this sentence into:

*"The input for the radiative transfer simulations was adapted to campaign-specific conditions."*

l.23 : are these ≪ snow properties ≫ used later on ?

Yes, these measurements were partly used in the simulations. This is mentioned in Section 2.3, where the snow pack radiative transfer model is introduced.

**p.5**

l.5 : **is** available

Changed as suggested.

l.10 : consider adding some information about the thermodynamical profiles measured during the flights, if actually used further

Yes, the humidity profiles are used to explain the heating rate profiles and should be added. We included the figure as a second panel to Figure 2, which shows the BC profiles. The discussion of the atmospheric profiles was extended to:

*"Fig. 1b shows the profiles of relative humidity, used for the simulations. PAMARCMiP was characterized by rather dry air. Only in the boundary layer, an average humidity up to 60 % was observed often linked to boundary layer clouds. ACLOUD and ARCTAS showed a higher relative humidity in higher altitude of up to 6 km, which indicates the influence of higher level clouds."*

[Figure]

Figure 1. Mean profiles of atmospheric BC particle mass concentration (a) and relative humidity (b) averaged for each the three campaigns (ACLOUD, ARCTAS and PAMARCMiP) as used for the radiative transfer simulations. Horizontal bars indicate the standard deviation. The positions of the two implemented cloud layers (blue shaded area) are marked.

l.13 : url for libRadtran download should be provided here or in the Data availability section

> The URL was added as suggested.

l.15 : is reference to « Evans 1998 » relevant here ?

> Thanks for identifying this mistake. The reference was removed.

l.15 : precise that this assumes a plane-parallel atmosphere

> In the revised manuscript we added a short justification of the assumption of a plan-parallel atmosphere:
>
> *"For the calculations, a plane-parallel atmosphere was assumed, which is justified for the Arctic conditions during the three campaigns. Using a pseudo-spherical geometry in libRadtran would change the broadband downward irradiance by less than 0.1 % (0.7 %) for a calculation with a SZA of 60° (75°)."*

l.19 : mention explicitly humidity (or water vapor)

> Added as suggested

**p.6**

l.1 : « adjusted » is unclear. Do you mean that profiles from the mid-campaign were used ?

> Yes, we used values representative for the campaign, which was the mid-campaign period. The sentence was changed into:
>
> *"Corresponding to the campaign average BC profiles, the range of the SZA values was set to values representing the campaign conditions (see Table 2)."*

*"The standard profiles were adapted to observations from radio soundings near the airborne observations or dropsondes released during the flights and represent the middle of the individual campaign periods."*

l.7 : where do the cloud optical properties come from ?

Yes, this important information was completely missing. In the revised manuscript ee added:

*"Optical properties of the liquid cloud were calculated from Mie-Theory, while the ice crystal optical properties are based on (Fu, 2007)."*

l.9 : can you provide optical thickness values ?

Sure, we should add the optical thickness and did so in the revision:

*"The assumed cloud properties correspond to a cloud optical thickness of 15 for the water cloud and 0.2 for the thin ice cloud."*

**p.7**

l.2 : provide url for TARTES

We added a web link.

l.6 : The reference provided is not about delta-Eddington approximation. Prefer Joseph et al. (1976)

Thanks for identifying this mistake. We changed the reference as suggested:

*"To solve the radiative transfer equation, the delta-Eddington approximation (Joseph et al.,1977) is used."*

l.8 : SSA should not be italic (throughout the text)

Changed as suggested.

l.9 : there are two shape parameters (B and g). Please provide the values used.

In the revised manuscript, these parameters were added.

*"Furthermore, the specific values of the so-called absorption enhancement parameter $B= 1.6$ and the geometric asymmetry factor $g^G= 0.85$ were applied."*

l.12 : ot → to

Changed as suggested.

l.13 : another important point is that impurities are assumed to be Rayleigh scatterers

We added this important fact to the revised manuscript.

*"The impurities are externally mixed and assumed to interact by Rayleigh scattering."*

l.23 : please provide some references for the SSA values assumed

The values are based on our measurements during PASCAL and PAMARCMiP. We added this in the revised manuscript.

*"The default values of snow density and SSA were based on measurements during PASCAL and PAMARCMiP and were set to 300 kg m$^{-3}$ and 20 m$^2$kg$^{-1}$, respectively."*

l.33 : SSA for fresh snow could be larger

Yes, we agree, that fresh snow can have larger values of SSA. However, we chose this value based on the measurements during PASCAL (ACLOUD), where in late spring those values were reported. In the revised manuscript we clarified that the assumption is based on measurements.

*"The top layer was assumed to be of fresh and clean snow with …. representing measurements from the PASCAL campaign."*

**p.8**

l.5 : no, spectral albedo does not depend on the spectral distribution of irradiance. Broadband albedo does

Sure, this only refers to broadband albedo. We removed "spectral" in the revised manuscript.

l.7 : ≪ shifts ≫ suggests a conversion of some wavelengths to some others. Maybe say ≪ filters/absorbs longer wavelengths so that the downward irradiance spectrum is shifted towards shorter wavelengths ≫

We reformulated this sentence to:

*"The transition from cloudy to cloudless atmospheric conditions increases the direct-to-global ratio ($f_{dir/glo}$) and the contribution of short wavelengths to the broadband downward irradiance (Warren, 1982)."*

l.9-10 : for which snowpack ?

We added this information in the revised manuscript:

*"For example, simulations with TARTES assuming cloudless and cloudy conditions changed the broadband snow surface albedo from about 0.8 to 0.9 for a SZA of 60° and a snow pack (no impurities) characterized by SSA= 20 m$^2$ kg$^{-1}$."*

**p.9**

Figure 2 : ≪ adjusted ≫ parameters is not clear. Do you mean that they can vary ? Maybe specify that the procedure is done at high spectral resolution so that the figure holds for a single wavelength. Consider adding a title with TARTES/libRadtran (or SNOW/ATMOSPHERE) on top of the colored boxes.

Thanks for the hint. We adjusted the scheme and figure caption:

[Figure]

*Figure 2. Schematics of the coupling of TARTES (gray box) and libRadtran (blue box) by exchanging the spectral surface albedo and the direct-to-global ratio. The list of varied parameters addresses the variables which were changed between the different realizations. Only the iterated parameters $f_{dir/glo}$ and $\alpha_\lambda$ are adjusted within an individual iteration cycle.*

l.4 : surface radiative effect is not clear (radiative forcing ?)

In the revised manuscript, we distinguish between the surface radiative forcing, which has a well-established definition and the BC radiative effect on the vertical heating rate profiles. We did go through the entire manuscript and exchanged "effect" by "forcing" wherever it refers to the surface radiative forcing. We hope that this makes it more clear, which quantity BC affects in the different discussions.

l.7 : specify what the default cases are when snowpack is considered (what BC in atmosphere ?) or atmosphere is considered (what BC in snow ?)

Yes, this was not fully described. Now we added the definition of the clean reference cases:

*"For the separated forcings, $F_{net,clean}$ refers to either a clean atmosphere or a clean snow layer, while the other part does consider BC particles. The default case of a clean atmosphere uses a BC mass concentration in the snow layer of 5 ng $g^{-1}$. Vice versa, the default case of a clean snow layer assumed the atmospheric BC profile of the ACLOUD campaign. For $\Delta F_{tot}$, the clean reference assumed both a pristine atmosphere and pristine snow layer."*

l.11 : can ? Should be ≪ does ≫ ?

Changed as suggested.

**p.10**

l.5-7 : the details about the vertical resolution of both radiative transfer codes should be given earlier in the presentation of the models configurations.

> As suggested, we moved this description into the model configuration section.

l.13 : daily means may hide much larger instantaneous values which are very relevant both for snow metamorphism and atmospheric dynamics. Adding the max values on the subsequent plots would be very useful

> We decided to analyze daily mean values to have a better quantification of the total daily effect with respect to the surface energy budget. Heating rates are given in K per "day". Showing the maximum values can be misleading, as the reader may conclude from the unit, that these values are relevant for the complete day. This, we aim to avoid, although we are aware that the maximum heating rates can be higher. Adding the maximum values is also no option as the range of the scale would need to be enlarged and reduce the visibility of the daily mean values.

**p.11**

l.2 : should be HR_BC ?

> Here we mean heating rates in general including the total heating rates and the efficient heating rate of BC. In the revised version we listed all calculated quantities.

l.6 : what kind of dependence ?

> We changed this sentence into:

> *"The reduction of the snow surface albedo by BC impurities depends on the snow grain size."*

l.6 : ≪ respectively ≫ is awkward

> We deleted the bracket.

l.15 : I don't see any zoom of the Figure 4, but definitely this would be useful

We are sorry for the confusion. We included the wrong image file in our first version. It is updated as follows:

[Figure]

**Figure 4.** Spectral surface albedo of snow for cloudless conditions and a SZA of 60° for different SSA and BC particle mass concentrations. The inlay shows an enlargement of the spectral albedo between 350 and 700 nm.

l.31 : please specify that this is the impact of BC on the broadband albedo. Other optical quantities might be much more altered

    We reworded this sentence as suggested:

    *"Therefore, for Arctic conditions, the impact of BC impurities on the broadband snow albedo is of minor importance, compared to the impact of modifying the snow grain size."*

**p.12**

Table 4 : particle

    Changed as suggested.

l.3 : the distinction between titles 3.1.1 and 3.1.2 is not obvious. Say radiative forcing ?

    In the revised manuscript we changed the terminology and used "forcing" for the instantaneous effect of BC on the surface radiative energy budget.

l.5 : twice ≪ effect ≫

    Thanks for identifying this typo. We corrected this sentence.

l.5 : **first** calculated… (because standard is with daily cycle). Note also that sometimes the past is used, sometimes the present. It'd be worth homogenizing this.

    We changed this sentence into:

*"To quantify these radiative effects, $\Delta F_{snow}$ was first calculated for a fixed solar zenith angle of 60° only. A typical Arctic range of BC particle mass concentrations in snow and SSA values assuming the ACLOUD atmospheric conditions were applied."*

**p.13**

l.8 : are **then** analyzed

Changed as suggested.

**p.14**

l.1 : how much in %? Using relative contributions rather than absolute forcing may be instructive to compare campaigns. This holds also when evaluating the contribution of clouds. Of course they shade the surface, but how does the relative forcing of BC vary ?

We tried to avoid using relative numbers as these might be misleading. Even small absolute effects may be large in relative numbers but still not relevant for the radiative energy budget. There is also no reference to what the radiative forcing can be compared to. In clean cases the forcing is zero, which makes it difficult to calculate relative numbers. The relative effect of clouds can be easily read from Figure 6 and does not need to be given in % in the text to our opinion.

**p.16**

l.5 : of by

Changed as suggested

l.15-16 : already said in the introduction

We removed this sentence in the revised manuscript.

l.19 : remove ≪ were applied ≫

Changed as suggested

l.32 : ≪ is less pronounced … significantly ≫

Changed as suggested

**p.17**

l.2 : one order of

Changed as suggested

l.15 : slight

Changed as suggested.

l.27 : ≪ to access ≫ is awkward

This sentence was changed due to another comment.

l.32 : ≪ transmittance ≫ is unclear. Do you mean irradiance with respect to surface irradiance ? Then relative illumination or relative irradiance is better.

Following an earlier comment, we included profiles of the transmissivity in the revised manuscript and extended this discussion.

l.33 : what is ≪ quickly ≫ for a heating rate decrease?

This sentence was changed into:

*"For all cases, the total heating rate rapidly decreases by one magnitude within the first 10 cm of depth."*

**p.19**

l.2 : I'm surprised not to see the shading of the lower layers by BC in the topmost layers. Did you observe that below?

We think that the shading is only hard to identify in the profiles of heating rates. More suited are the transmissivities which are now included in the revised Figure 8 (see comments above). Only for the multi-layer scenario, the shading is obvious in the lowest snow layer. Here, the heating rates decrease toward zero, while the top layer (same amount of BC) shows a non-zero heating rate. With the revised Figure 8, this is also visible in the transmissivities.

l.12 : = 0 or ≈ 0 ?≫

Changed as suggested.

l.17 : I think the conversion from a contribution to a snow heating rate into a metamorphism rate is not that straightforward, especailly with daily means. Providing snow physics references would be helpful to strengthen your conclusions

Yes, our conclusion was not well justified. In the revised manuscript, we extended the discussion and compared to results for alpine snow.

*"Therefore, in Arctic conditions the snow grain size typically plays a larger role than the concentration of BC particles embedded in snow. To estimate if BC particles can accelerate the snow metamorphism, coupled snow physical models need to be applied (Tuzet et al., 2017). However, compared to the results reported by Tuzet et al. (2017) who studied alpine snow with at least a magnitude higher BC mass concentrations, for Arctic conditions it is likely, that the self-amplification of the snow metamorphism is dominated the reduction of the surface albedo."*

l.33 : were used **in** the

Changed as suggested.

**p.20**

l.3 : ≪ other parameters ≫ is awkward. Please clarify. Maybe mention reference/unpolluted conditions

The sentence was changed to:

*"For the heating rate profiles, the effective contribution of BC particles to the total heating rates was derived and compared to further atmospheric and snow parameters also leading to a warming or cooling (e.g., water vapor, clouds, snow grain size)."*

l.5 : again, ≪ local ≫ is unclear

> As explained above, "local" refers to a local instantaneous effect which cannot be used for the entire Arctic. The changes the sentence to:
>
> *"The simulations suggest, that for the specific Arctic cases investigated in our study, the radiative forcing of BC is small compared to the radiative impact of other parameters (water vapor, clouds, snow grain size)."*

l.6 : why ≪ therefore ≫ ?

> We deleted "therefore".

l.6 : shows

> Changed as suggested.

l.7 : ≪ driver ≫ means that its variability controls the variability of the heating rates. Is that the case (then it should be detailed) ? You could have varying BC for constant water vapour, then the variations of the heating rates would be driven by BC.

> Yes, our conclusion was not expressed precisely and might be misleading. We changed the sentence to:
>
> *"In cloudless conditions, the absorption by atmospheric water vapor shows a much stronger contribution to the atmospheric heating rates than the radiative effect of BC particles."*

l.15 : lapse-rate feedback refers to the response of the atmosphere to a surface temperature change. In terms of vertical gradient of temperature. I'm not sure you really mean this here (as a feedback).

> Thanks again! We remove "feedback" as we can only assume what happens to the lapse rate without all feedback mechanisms.

l.16-17 : again, what is ≪ small ≫ ? can you provide elements to support the fact that 0.1 K day-1 cannot change atmospheric stability ?

> Of course, also 0.1 K numerically changed the atmospheric stability. However, the effect is about two magnitudes smaller than calculated for polluted regions (8 K day$^{-1}$ reported by Wendisch et al. 2007). In these polluted cases, changes of the temperature profile by advection, radiative cooling might be slower than the heating by BC particles. But the rate of 0.1 K per day is too slow compared to other processes. We changed the sentence to make this more clear.
>
> *"For example, studies investigating strong pollution conditions in northern India or China reported on BC heating rates in the atmosphere larger than 2 K day$^{-1}$, which may significantly influence the lapse rate and the atmospheric stability (Tripathi et al., 2007; Wendisch et al., 2008). For the rather pristine Arctic, this study showed significantly lower daily mean BC heating rates of maximum 0.1 K day$^{-1}$, which have not the potential to significantly modify the atmospheric stability."*

l.17-18 : this could be moved to the introduction, that the study focuses on remote Arctic locations, not on locally polluted areas.

To make this more clear, we adjusted the introduction. But we like to keep this sentence also for the discussion in the conclusion section.

*"The area of interest is the remote sea ice covered Arctic Ocean in the vicinity of Spitsbergen, northern Greenland and northern Alaska typically not affected by local pollution."*

l.33 : these **two** cloud

Changed as suggested.

**p.21**

l.4 : ≪ Atlantic Arctic ≫ should be emphasized in the introduction

Atlantic Arctic was not correct, as we also use data from ARCTAS (Alaska). Therefore, we changed the abstract to:

*"The area of interest is the remote sea ice covered Arctic Ocean at latitudes of Spitsbergen, northern Greenland and northern Alaska typically not affected by local pollution."*

l.5 : cooling is at the surface, please clarify

Yes, this effect refers to the surface warming/cooling. We added "surface" in the revised manuscript.

l.10 : some elements should be provided about other types of impurities which may eventually be more critical than BC in the Arctic

Based on another comment, we extended the discussion with:

*"Furthermore, BC particles are not the only light absorbing impurities, which are transported into the Arctic. The relevance of dust particles and micro-organisms is currently subject of the scientific discussion and may exceed the effect of BC particles (Kylling et al., 2018, Skiles et al., 2018)."*

---

## Author Comment (AC2) · 15 Jun 2020

**Reply to Reviewer #2:**

We gratefully thank the reviewer for the detailed review and her/his valuable suggestions to improve the manuscript. Detailed replies on the reviewer's comments are given below. Our replies are given written with indention. Citations from the revised manuscript are given in italic and quotation marks.

Page 3, caption table 1: There are various ways of defining BC. Please include a reference to for example Petzold et al. (2013) to clearly define your use of BC. Also please mention how EC values compare with BC values.

Thanks to the reviewer to bring this up. We are aware that the definition of BC in literature is not consistent. However, Petzold et al., 2013 provided an excellent overview on that topic. Since the terminology depends on the different measurement techniques, we added the applied measurement method in Table 1.

Table 1. Typical values of the black carbon mass concentration in snow pack observed in different regions and seasons in the Arctic. Note, that Pedersen et al. (2015) and Forsström et al. (2013) derived the mass concentration of elemental carbon applying a thermal-optical measurement method.

| Location           | Season      | BC mass concentration (ng $g^{-1}$ ) | Method              | Source                  |
|--------------------|-------------|--------------------------------------|---------------------|-------------------------|
| Svalbard region    | March/April | 13                                   | filter transmission | Doherty et al. (2010)   |
| Arctic Ocean snow  | Spring      | 7                                    | filter transmission | Doherty et al. (2010)   |
| Arctic Ocean snow  | Summer      | 8                                    | filter transmission | Doherty et al. (2010)   |
| Northern Norway    | May         | 21                                   | filter transmission | Doherty et al. (2010)   |
| Central Greenland  | Summer      | 3                                    | filter transmission | Doherty et al. (2010)   |
| Svalbard region    | March/April | 11 - 14                              | thermal-optical     | Forsström et al. (2013) |
| Corbel, Ny-Ålesund | March       | 21                                   | thermal-optical     | Pedersen et al. (2015)  |
| Barrow             | April       | 5                                    | thermal-optical     | Pedersen et al. (2015)  |
| Ramfjorden, Tromsø | April       | 13                                   | thermal-optical     | Pedersen et al. (2015)  |
| Valhall, Tromsø    | April       | 137                                  | thermal-optical     | Pedersen et al. (2015)  |
| Fram Strait        | April       | 22                                   | thermal-optical     | Pedersen et al. (2015)  |

Further, we cited Petzold et al. (2013) in the introduction:

"Black carbon (BC) aerosol particles, which mostly originate from incomplete combustion of organic material (Bond et al., 2013; Petzold et al., 2013), absorb and scatter solar radiation in the visible wavelength range and, therefore, influence the atmospheric solar radiative energy budget."

and revised the manuscript accordingly:

"The numbers given in Table 1 were derived from different measurement methods. More precisely, thermal-optical techniques were applied in Forsström et al. (2013) and Pedersen et al. (2015) provide the elemental carbon (EC) mass concentration, while filter transmission methods result in BC concentrations (Doherty et al., 2010). As a consequence of the different measurement methods, the ratio of the BC to EC concentration in snow can reach values of 1.3 as reported by Douet al. (2017). A full discussion of the EC/BC terminology can be found in Petzold et al. (2013)."

Page 4, lines 17-19: Sentence is unclear. Please reformulate.

We rephrased the sentence:

"In this paper, measurements from the PAMARCMIP 2018 observations conducted from 10 March to 8 April 2018 were analyzed. The research flights, starting from Station Nord/Greenland, were performed above the sea ice in the Arctic ocean north of Station Nord and the Fram Strait."

Page 5, line 15: The reference to Evans (1998) and the SHDOM code appears to be out of place. Should it be Stamnes et al. (1988) instead?

We cite Stamnes et al., (2000) here which explicitly refers to DISORT2.0 as also indicated in Mayer and Kylling (2005):

"As a solver for the radiative transfer equation, the Discrete Ordinate Radiative Transfer solver (DISORT) 2 (Stamnes et al., 2000) routine running with 16 streams was chosen."

Page 5, lines 12-18: 1. How many streams was used for DISORT? 2. The solar zenith angle is large for all regions considered. Did you make any spherical corrections? If not, why not, and how do you expect this to affect your results? 3. What was the vertical resolution of your model atmosphere?

The number of streams (16) is given in the sentence before. Further, the reviewer raises a good question concerning the plane-parallel assumption we applied here. For testing the effect, we compared pseudo-spherical and plane-parallel for SZA ranging between 60 - 75° giving an uncertainty of less than 0.7% in downward irradiance. We added:

"For the calculations, a plane-parallel atmosphere was assumed, which is justified for the Arctic conditions during the three campaigns. Using a pseudo-spherical geometry in libRadtran would change the broadband downward irradiance by less than 0.1 % (0.7 %) for calculations with SZA =  $60^{\circ}$  (75°). The vertical resolution of the simulated irradiances was adjusted to the measured BC profiles, ranging between 100 m and 1 km."

Page 6, Fig 1: The profiles shown are averages. Please also include the standard deviation (or other measure of variability) of the profiles to give an idea of how the profiles varied for the different campaigns.

Thanks for the suggestion. We added the standard deviation in Figure 1 and included profiles of the relative humidity in a second panel as suggested by the other reviewer.

Figure 1. Mean profiles of atmospheric BC particle mass concentration (a) and relative humidity (b) averaged for each the three campaigns (ACLOUD, ARCTAS and PAMARCMiP) as used for the radiative transfer simulations. Horizontal bars indicate the standard deviation. The positions of the two implemented cloud layers (blue shaded area) are marked.

Page 7, line 2: In the snow a two-stream model is used. Presumably more streams were used for the atmospheric radiative transfer. Why is it sufficient to use only two streams in the snow pack?

The number of streams is related to the number of angles where the radiance is calculated. For up- and downward irradiance calculations often two-stream models are applied. In particular, for the radiative transfer simulations snow models apply the two-stream approximation. Dang et al. (2019) compared the DISORT calculation using 16 streams as benchmark with three two-stream models to identify the uncertainty of albedo simulations. Figure 2 from Dang et al. (2019) shows the simulated snow albedo for the tested models, illustrating the sufficient accuracy of the two-stream approximation.

Figure 2 from Dang et al. (2019)

They conclude: "Compared with a 16-stream benchmark model, the errors in snow visible albedo for a direct-incident beam from all three two-stream models are small (<±0.005) and in-crease as snow shallows, especially for aged snow. The errors in direct near-infrared (near-IR) albedo are small (<±0.005) for solar zenith angles  $\theta$  <75°, and increase as  $\theta$  increases."

We are aware that for SZA >  $75^{\circ}$  the uncertainty by using the two-streams approximation might be higher than 0.005.

Dang, C., Zender, C. S., and Flanner, M. G.: Intercomparison and improvement of two-stream shortwave radiative transfer schemes in Earth system models for a unified treatment of cryospheric surfaces, The Cryosphere, 13, 2325–2343, https://doi.org/10.5194/tc-13-2325-2019, 2019.

line 6: Stamnes et al. (1988) is not a reference for the delta-Eddington approximation. Maybe rather cite Joseph et al. (1976)?

Thanks for identifying this mistake. We changed the reference as suggested:

"To solve the radiative transfer equation, the delta-Eddington approximation (Joseph et al., 1977) is used."

Page 7, line 14: In the snow the BC optical properties are from Bond et al. (2013) while in the atmosphere they are from Hess et al. (1998). Hence, the BC particles are different in the atmosphere and the snow. What is the rationale behind this choice other than what is available in the models used?

The refractive index of BC can vary a lot and is reported differently in various publications. Bond et al., 2013 writes exemplarily: "A variety of values for the refractive index of BC has been used in global climate models including the OPAC value of 1.74 +- 0.44i [Hess et al., 1998]." In this study we decided to use the data of BC optical properties as proposed by the two separate models for radiative transfer simulations in snow and in atmosphere, respectively.

Page 8, Table 3: Should the first row in the table be named "Thickness" instead of "Depth"?

In literature we found both terms. However, we stick to the term "depth". It makes sense following the explanation on https://wikidiff.com/depth/thickness: "As nouns the difference between depth and thickness is that depth is the vertical distance below a surface; the degree to which something is deep while thickness is (uncountable) the property of being thick (in dimension)."

Page 8, lines 20-21: The sentence "This procedure is repeated until the deviation between previous (step n) and revised surface albedo decrease below 1 %" is unclear. Please reformulate.

The sentence is revised as follows:

"This procedure was repeated until the deviation of the surface albedo calculated in the previous step (n) and calculated in the revised step (n+1) decreases below 1 %."

Page 10, Fig. 3: May it be concluded from the plot that the iteration procedure has no impact on the surface albedo in the wavelength region where BC absorbs?

The reviewer is right. The coupling is of minor importance for the surface snow albedo in the wavelength range where BC in snow absorbs. In TARTES, the calculation of the snow albedo requires the direct-to-global ratio as boundary condition. The difference of the calculated snow albedo from one iteration step to the next depends strongly on the change of the direct-to-global ratio. As indicated in Fig. 3, the initial step assumes a ratio of 0, which is more appropriate for the visible spectral range than for the nearinfrared. For cloudless conditions the direct-to-global ratio is almost one in the nearinfrared. Therefore, the largest effect of coupling is observed in the near-infrared spectral range.

We added:

"The assumption of a pure diffuse illumination in the initial run caused no significant difference of the calculated visible snow albedo to the first and second iteration step. In contrast, the iterated direct-to-global ratio adjusts the snow albedo in the near-infrared, because the direct fraction is quickly approaching unity in this spectral range."

Page 11, lines 1-2: The upward and downward irradiances were averaged and from these the averaged heating rates were calculated. This appears as a rather unusual and unphysical approach. Would it not be more appropriate to calculate the instantaneous heating rates and then average these?

In this study we were focused on the daily averaged heating rates. The averaging gives exactly what the unit of heating rates, K/day, is expressing. From the mathematical point of view, there is no difference between the temporal averaging of the irradiances and calculating the mean heating rate out of it, or averaging the temporal resolved heating rates over the day.

Since the numerator is a linear term and the arithmetic mean has linear correlations, the results will not change when swapping the order of operation. We try to illustrate that by the following equation:

$$\overline{HR(z)} = \frac{\frac{1}{n}\sum_{i=1}^{n} (F_{net}z_t, i - F_{net}z_b, i)}{\rho(z)c_p(z_t - z_b)} = \frac{\frac{1}{n}\sum_{i=1}^{n} (F_{net}z_t, i) - \frac{1}{n}\sum_{i=1}^{n} (F_{net}z_b, i)}{\rho(z)c_p(z_t - z_b)} = \frac{1}{n}\sum_{i=1}^{n} HR(z), i$$

Page 11, line 15: The enlargement of Fig 4. seems to be missing. As Fig. 4 is, it does not make sense to have many overlapping lines. Please provide a zoom in of the visible wavelength region (lambda

---

## Author Response (AR1)

**Reply to Quentin Libois (Reviewer #1):**

We gratefully thank the reviewer for the detailed review and the numerous larger and smaller suggestions. The comment guided us easily to improve the manuscript. We would like to highlight the efforts of the reviewer, for reading the manuscript very carefully and identifying many typos.

Detailed replies on the reviewer's comments are given below. Our replies are given written with indention. Citations from the revised manuscript are given in italic and quotation marks.

**General comments**

This study aims at estimating the radiative impact of black carbon (BC) particles suspended in the atmosphere and contained in the snowpack in the Arctic. It simultaneously and consistently computes the radiative forcing of BC in both the snowpack and the atmosphere. To this end it couples an atmospheric and a snow radiative transfer model. The BC atmospheric concentrations are taken from three aircraft campaigns that explored various atmospheric conditions, from early spring to summer. A variety of radiative transfer simulations are performed, where snow properties and BC mass concentrations are varied to cover the range of Arctic conditions reported in the literature. The main conclusion is that the radiative impact of BC is marginal in typical Arctic conditions, amounting to about a few percent of the total heating rates and to less than 1 W m-2 in terms of surface forcing. The authors also point a competition between shading of the surface by atmospheric BC that counteracts the warming effect of BC in snow. The impact of clouds is investigated, also showing complex interactions, where depending on their altitude and optical thickness, clouds can either enhance the effect of BC through multiple scattering, or reduce it by shading. In any case, the authors highlight that other drivers of the Arctic energy budget are more significant than BC, such as absorption by water vapour, snow metamorphism and clouds.

The topic of the study is relevant to ACP because it combines numerical simulations and field observations to provide a geophysical analysis. The paper is well written and easy to follow. There is much relevant physical insight and the conclusions are drawn rigorously from the computations. The findings are not a breakthrough but they have the merit to provide a selfstanding investigation of the total BC impact in Arctic conditions, where previous studies have either focused on the atmosphere or in the snow. This is probably the greatest added value compared to previous work. We may regret the lack of field data for the snow. Likewise, the fact that only offline radiative computations are performed precludes a rigorous quantification of the impacts on atmospheric dynamics and snow evolution. As a consequence, the numerous conclusions on the impact of BC with a dynamic perspective appear quite weak and should be better motivated with appropriate references. Practically, data from aircraft campaigns are only used to derive average profiles of temperature, humidity (in a manner that should be more detailed) and BC, but snow properties are chosen based on other studies and more as varying parameters. This is not an inappropriate approach but this makes the importance of novel data quite limited in this work. Based on the comments above, I recommend this paper be published after the corrections suggested below are tackled.

Again, we thank the reviewer for summarizing the open issues of the original manuscript. The replies on the following specific comments hopefully consider also the general concerns raised by the reviewer.

**Specific comments**

1) It is clear that the study focuses on BC and the conclusion is that BC is not so critical with the amounts currently observed in the remote Arctic. However, recently there have been plenty of studies clearly showing altered surface albedos because of light absorbing impurities. The latter could then be dust, micro-organisms or anything else. It might be worth insisting that you only deal with BC, which is one amongst many others light absorbing species, so that the conclusion should not be over-interpreted as  $\ll$  there is no impact of impurities in the Arctic  $\gg$ . Likewise, the geopgraphical area to which the work is relevant should be better identified.

We agree with the reviewer, that the estimates of the BC radiative effect calculated in our study cannot be generalized. Other impurities might give a more significant signal. Also we restricted our analyses to snow on sea ice. The Effects of BC might accumulate as BC particles accumulate when snow melts and bare sea ice is left. Ever more important is BC on glaciers where the accumulation does last more than the 1-3 years before sea ice typically melts. In the revised manuscript we emphasized the limitations of our calculations at several instances:

"For the conditions over the Arctic Ocean analyzed in the simulations, it is found, that..."

"This study analyzed the instantaneous solar radiative effect at the surface of Arctic BC particles (suspended in the atmosphere and embedded in the snow pack) over the sea ice covered Arctic Ocean."

"It needs to be considered, that this picture might change if the accumulation of BC particles is more efficient than it is over the snow covered Arctic sea ice, where the sea ice and snow pack does not last more than one to three years. Accumulation of BC on e.g. the Greenlandic glaciers will amplify the radiative forcing on a local scale. Furthermore, BC particles are not the only light absorbing impurities, which are transported into the Arctic. The relevance of dust particles and micro-organisms is currently subject of the scientific discussion and may exceed the effect of BC particles (Kylling et al., 2018, Skiles et al., 2018)."

2) The paper focuses on energy budgets (of the atmosphere and snow). Although the impact of BC on these budgets is very limited, BC strongly impacts the light penetration depth in snow, or equivalently snow transmittance. For instance, if a 20 cm snow layer in the Arctic has a transmittance of 1 %, adding BC may decrease this value down to 0.5 %. This is nothing for the snow budget, buth this makes a huge difference for the amount of energy transmitted. This will for instance be critical for photosynthesis within or under the snowpack. Maybe this should be mentioned somewhere so that again readers don't think  $\ll$  BC does not matter  $\gg$ . The paper by Tuzet et al., (2019) may be a useful reference for that.

Thanks for pointing at this relevant aspect which we did not consider so far. Indeed, our simulations show a significant decrease of transmissivity below the snow layer. For the homogeneous snow layer, the transmissivity in 20 cm depth for the unpolluted case is about 0.3, while adding a BC concentration of 5 ng g4 reduces the transmissivity to almost 0.2. This obviously may have an impact on the radiative processes in and below the sea ice. In the revised manuscript, we added an additional panel to Figure 8 showing the transmissivity profile and added a short discussion.

Fig. 8a: Transmissivity profiles of solar radiation within the snow pack for three single layers and one multi-layer case assuming ACLOUD conditions.

"Figure 8a shows the transmissivity profiles of solar radiation within the snow pack. The homogeneous single layer reference case without BC particles (SSA =  $20 \text{ m}^2 \text{ kg}^{-1}$ ) illustrates the general decrease of transmissivity, which is reduced to 0.3 in 20 cm snow depth. Adding a typical Arctic BC concentration of 5 ng g-1 reduces the transmissivity to almost 0.2. This obviously may have an impact on the radiative processes below the snow pack, in and below the sea ice as discussed by, e.g., Tuzet et al (2019) and Marks and King (2014). The inhomogeneous multi-layer case shows in general lower transmissivities due to the enhanced reflection of the smaller snow grains at top of the layer (SSA =  $60 \text{ m}^2 \text{ kg}^{-1}$  down to 5 cm depth) but also indicates a significant dimming effect of the BC particles."

3) The iterative coupling between libRadtran and TARTES is a first valuable step towards consistent radiative transfer simulations. I can only encourage the authors to fully incorporate the scattering snowpack in libRadtran for their future work. This can be done simply by providing the single scattering properties computed by TARTES to create new « atmospheric » layers in libRadtran which would be extremely thin. Such strategy would avoid the iterative coupling and be overall more elegant. See for instance Blanchet and List (1987) for a very similar study.

We agree, a full coupling of both models is the final goal for further studies combining atmosphere and snow radiative transfer. For this study we first aimed to test, if the coupling is possible in general and how large the interaction is. As the iterative coupling shows a very quick convergence, we concluded that this iterative coupling is sufficient for this study. However, for future studies, we will consider the advice of the reviewer.

4) The evaluation of BC contribution to heating rates or total absorption is sometimes misleading. The authors often conclude that BC contribution being a few percents its impact is negligible. However think in terms of CO2 forcing, where a few W m-2 (in addition to hundreds of W m-2) can fully change the face of the Earth. I simply mean that it is hard to conclude that

a few percents perturbation of the energy budget due to BC is insignificant. Be more cautious in the conclusions, unless you have strong and better argumented reasons to think that it is indeed negligible.

We agree that also only a few W m-2 radiative effect can be significant in terms of the total Arctic wide energy budget. As our study is based on three campaigns, the Arctic wide absolute BC radiative effect (or even forcing) cannot be assed. That's why comparing the W m-2 of our study to the C02 forcing would be misleading. Also because the BC radiative effect is a local instantaneous radiative effect, while the climate effect would include all relevant feedbacks. Therefore, we always compared the BC radiative effect to other radiative effects of other properties, e.g., atmospheric water vapor, clouds, snow grain size. At no point in the manuscript we claim, that the BC effect is negligible for the total energy budget. Our conclusion is, that compared to BC radiative effects, other drivers are more important and these other parameters first need to be constrained more precisely to improve e.g., Arctic climate models.

In the revised manuscript, we tried to check all our conclusions and adjusted the wording if needed:

"The magnitude of solar radiative effects (cooling or warming) of black carbon (BC) particles embedded in the Arctic atmosphere and surface snow layer were quantified on the basis of case studies."

*"However, in other Arctic regions characterized by higher atmospheric BC particle concentrations due to local fires, e.g., northern Siberia, a stronger impact can be expected."*

"These results indicate, that the microphysical properties of the snow pack (mainly snow grain sizes) are more important drivers for the degree/strength of the snow metamorphism. It needs to be considered, that this picture might change if the accumulation of BC particles is more efficient..."

5) The paper somehow lacks a bit of discussion, where the limits of the study and recommendations for future work would be provided. In particular, the importance of BC in locations where it is much more concentrated could be discussed. The representativity of the BC atmospheric profiles used as well. The use of daily averages to asses a radiative impact may not be relevant (maximum values matter as well). The link with snow metamorphism is only qualitative why models allow to explicitly simulate the impact of these heating profiles of metamorphism, etc. All these points should be brought to the reader and further investigated in future work, if not already further discussed in the present paper.

We agree, that the discussion of our results was lacking in detail. In the revised manuscript we tried to address all issues raised here by the reviewer. As all of the single issues are listed in the technical correction, we did not explicitly reply here and refer to the replies given below.

**Technical corrections**

title : would **forcing** be more appropriate than « effects »? Consider also removing «layer»

For the title, we do not think that forcing is appropriate. We calculate the radiative effects (forcing) on the surface radiation budget but also the effect on heating rate profiles. To our understanding, the term "forcing" is linked to the energy budget only. So we would keep "radiative effect" in the title.

"layer" is removed.

p.1

I.1 : the abstract could be written using the present. More generally there is no consistent use of present or past in the manuscript. Some homogenization would be recommended.

We are sorry that we often struggle with the use of the correct tense. We tried to follow our experience publishing in Copernicus journals, where Copy-Editing mostly changes tense into past, when things are done in past. We tried again to homogenize the text and will hope for advice from the final copy-editing process.

I.2 : BC particles are not really  $\ll$  suspended  $\gg$  in the snowpack. They're rather embedded or contained. Consider changing this throughout the paper.

Changed to embedded throughout the paper

I.2 :  $\ll$  by  $\gg$   $\rightarrow$  using

Changed as suggested.

 $I.4: \ll$  interactions  $\gg$  is unclear. Maybe use multiple scattering or coupling

We kept "interactions" as multiple scattering is only one process which is considered when coupling the two models. E.g. also the change of direct to diffuse incoming radiation, which is not driven by multiple scattering alone, changes the radiative properties of the surface.

I.4 :  $\ll$  a snow layer  $\gg$  should be replaced by  $\ll$  a snow  $\gg$  because multi-layer snowpacks are explored. Maybe write  $\ll$  An atmospheric and a snow radiative...  $\gg$

Changed as suggested.

I.6 : this radiative effect is very dependent on the SZA chosen. Please clarify

We calculated daily mean values considering the diurnal change of the solar zenith angle. Sure, still the results depend on the location, time of year. We therefore added the minimum solar zenith angle and pointed out that the numbers give daily mean estimates of the BC radiative effects.

"For pristine early summer conditions (no atmospheric BC, minimum solar zenith angles of 55°) and a representative BC particle mass concentration of 5 ng g-1 in the surface snow layer, a positive daily mean solar radiative effect of +0.2 W m-2 was calculated for the surface radiative budget."

I.9 : counteracting  $\ll$  effect  $\gg$

Changed to:

"The total net surface radiative forcing combining the effects of BC embedded in the atmosphere and in the snow layer strongly depends on the snow optical properties (snow specific surface area and snow density)."

I.10 : technically snow density also impacts snow optical properties

Density was added as suggested.

 $I.10: \ll$  however  $\gg$  does not really oppose to anything

Changed as suggested.

I.12 : I think  $\ll$  ice  $\gg$  could be used instead of  $\ll$  ice water  $\gg$

Changed as suggested.

I.19 : absorbs, scatters

Thanks! changed as suggested.

L.24 : predominantly

Thanks! changed as suggested.

**p.2**

I.1 in higher  $\rightarrow$  to higher

Changed as suggested.

I.4-5 : using  $\ll$  nevertheless  $\gg$  and  $\ll$  still  $\gg$  in two consecutive sentences makes it difficult to follow

We changed the sentences to:

"In future, a strong intensification of the ship traffic in the Arctic Ocean and further polluting human activities are expected (Corbett et al., 2010). Still, the direct ..."

 $I.7: \ll of suspended \gg is awkward$

We corrected this typo.

I.9 : can be expected  $\rightarrow$  are observed

Changed as suggested.

I.14 : double  $\ll$  the  $\gg$

We corrected this typo.

I.20 : associated with  $\rightarrow$  , thus increasing the amount...

We changed the sentence to:

"The absorption effect can add to the warming of the atmosphere or the snow pack, when the BC particles are suspended either in the air or embedded in the snow. Furthermore, the BC particles may lead to a reduction of the snow surface albedo if the BC sediments on or into the snow pack (Sand et al. 2013)."

1.29: there is no  $\ll$  novel  $\gg$  feedback described here. BC is just shown to trigger the snow metamorphism feedback. There is actually a feedback because BC impact will be stronger for lower SSA, but this should be described here if this is what you actually mean.

Yes, we did not precisely distinguish booth feedback mechanisms. In the revised manuscript, we changed this into:

"As a further consequence, the absorption by BC particles supports the melting of snow and increases the snow grain size due to an enhanced snow metamorphism, leading to further reduction of the surface albedo. The increase of the snow grain size also feeds back to the absorption by BC particles, which is more efficient for larger snow grain sizes (Warren and Wiscombe, 1980)."

I.34 : warming

Changed as suggested.

I.34 :  $\ll$  the atmospheric layer containing BC  $\gg$

Changed as suggested.

**р.3**

I.4-10 : this paragraph is not very clear and could probably be removed

As we would like to keep this model aspect in the introduction, we did rewrite the paragraphs as follows to make the statements more clear.

Several regional and global climate models account for the opposite radiative effects of atmospheric BC particles and snow-embedded BC particles (Samset et al., 2014). However, estimates of the total net forcing rely on the accuracy of the distribution of the BC particles assumed in the particular model. Samset et al. (2014) compared 13 aerosol models from the AeroCom Phase II; all of them included BC. They found that modeled atmospheric BC concentrations often show a spread over more than one order of magnitude. In remote regions, dominated by long range transport, these models tend to overestimate the atmospheric BC particle mass concentrations compared to airborne observations. On the other hand, an underestimation of deposition rates induces a lower BC mass fraction in snow (Namazi et al., 2015). While this may introduce significant local and temporal uncertainties of the BC concentration and related radiative effects, long-term trends and mean multi-model results are representative for Arctic-wide observations (Sand et al., 2017)."

I.15: why only  $\ll$  local  $\gg$ ? Not clear whether this refers to local pollution or not

Our aim was to clarify, that our estimates are not general for the entire Arctic. As local obviously can be misleading, we changes the sentence into:

"On the basis of measured Arctic BC particle mass concentrations for spring and summer months, the instantaneous radiative forcing of BC particles embedded in the snow surface layer and in the atmosphere were quantified for specific cases."

I.16 : With

Changed as suggested.

**I.15 : « interactions » is not very appropriate**

We changed this sentence into:

*"With help of the coupled model, the interaction of radiative effects in the atmosphere and the snow pack was considered."*

I.5 : « relevance » is not well chosen  $\rightarrow$  contribution

We changed the last two sentences into:

"Vertical profiles of heating rates in the atmospheric and in the snow pack are presented for clean and polluted conditions. To estimate the impact of BC particles, effective heating rates are calculated by separating the BC radiative effect from the total heating rates."

 $\rm I.7:\ll$  setup  $\gg$  suggests there is some evolution from an initiation which is not the case.  $\ll$  Configuration  $\gg$  would be better.

Changed as suggested.

I.8 : change title to  $\ll$ BC profiles from aircraft campaigns  $\gg$

Changed as suggested.

I.9 : not clear what this  $\ll$  atmospheric  $\gg$  model is

We changed this sentence into:

"The input for the radiative transfer simulations was adapted to campaign-specific conditions."

I.23 : are these  $\ll$  snow properties  $\gg$  used later on ?

Yes, these measurements were partly used in the simulations. This is mentioned in Section 2.3, where the snow pack radiative transfer model is introduced.

**p.5**

I.5 : is available

Changed as suggested.

I.10 : consider adding some information about the thermodynamical profiles measured during the flights, if actually used further

Yes, the humidity profiles are used to explain the heating rate profiles and should be added. We included the figure as a second panel to Figure 2, which shows the BC profiles. The discussion of the atmospheric profiles was extended to:

"Fig. 1b shows the profiles of relative humidity, used for the simulations. PAMARCMiP was characterized by rather dry air. Only in the boundary layer, an average humidity up to 60 % was observed often linked to boundary layer clouds. ACLOUD and ARCTAS showed a higher relative humidity in higher altitude of up to 6 km, which indicates the influence of higher level clouds."

Figure 1. Mean profiles of atmospheric BC particle mass concentration (a) and relative humidity (b) averaged for each the three campaigns (ACLOUD, ARCTAS and PAMARCMiP) as used for the radiative transfer simulations. Horizontal bars indicate the standard deviation. The positions of the two implemented cloud layers (blue shaded area) are marked.

I.13 : url for libRadtran download should be provided here or in the Data availability section

The URL was added as suggested.

I.15 : is reference to  $\ll$  Evans 1998  $\gg$  relevant here ?

Thanks for identifying this mistake. The reference was removed.

I.15 : precise that this assumes a plane-parallel atmosphere

In the revised manuscript we added a short justification of the assumption of a planparallel atmosphere:

"For the calculations, a plane-parallel atmosphere was assumed, which is justified for the Arctic conditions during the three campaigns. Using a pseudo-spherical geometry in libRadtran would change the broadband downward irradiance by less than 0.1 % (0.7 %) for a calculation with a SZA of 60° (75°)."

I.19 : mention explicitly humidity (or water vapor)

Added as suggested

**p.6**

 $I.1: \ll$  adjusted  $\gg$  is unclear. Do you mean that profiles from the mid-campaign were used ?

Yes, we used values representative for the campaign, which was the mid-campaign period. The sentence was changed into:

"Corresponding to the campaign average BC profiles, the range of the SZA values was set to values representing the campaign conditions (see Table 2)."

"The standard profiles were adapted to observations from radio soundings near the airborne observations or dropsondes released during the flights and represent the middle of the individual campaign periods."

I.7 : where do the cloud optical properties come from ?

Yes, this important information was completely missing. In the revised manuscript ee added:

"Optical properties of the liquid cloud were calculated from Mie-Theory, while the ice crystal optical properties are based on (Fu, 2007)."

I.9 : can you provide optical thickness values ?

Sure, we should add the optical thickness and did so in the revision:

"The assumed cloud properties correspond to a cloud optical thickness of 15 for the water cloud and 0.2 for the thin ice cloud."

**p.7**

I.2 : provide url for TARTES

We added a web link.

I.6 : The reference provided is not about delta-Eddington approximation. Prefer Joseph et al. (1976)

Thanks for identifying this mistake. We changed the reference as suggested:

"To solve the radiative transfer equation, the delta-Eddington approximation (Joseph et al., 1977) is used."

I.8 : SSA should not be italic (throughout the text)

Changed as suggested.

I.9 : there are two shape parameters (B and g). Please provide the values used.

In the revised manuscript, these parameters were added.

*"Furthermore, the specific values of the so-called absorption enhancement parameter B= 1.6 and the geometric asymmetry factor*  $g^{G}$ *= 0.85 were applied."*

I.12 : ot  $\rightarrow$  to

Changed as suggested.

I.13 : another important point is that impurities are assumed to be Rayleigh scatterers

We added this important fact to the revised manuscript.

"The impurities are externally mixed and assumed to interact by Rayleigh scattering."

I.23 : please provide some references for the SSA values assumed

The values are based on our measurements during PASCAL and PAMARCMiP. We added this in the revised manuscript.

"The default values of snow density and SSA were based on measurements during PASCAL and PAMARCMiP and were set to 300 kg m-3 and 20 m2kg-1, respectively."

**I.33 : SSA for fresh snow could be larger**

Yes, we agree, that fresh snow can have larger values of SSA. However, we chose this value based on the measurements during PASCAL (ACLOUD), where in late spring those values were reported. In the revised manuscript we clarified that the assumption is based on measurements.

"The top layer was assumed to be of fresh and clean snow with .... representing measurements from the PASCAL campaign."

**p.8**

I.5 : no, spectral albedo does not depend on the spectral distribution of irradiance. Broadband albedo does

Sure, this only refers to broadband albedo. We removed "spectral" in the revised manuscript.

 $\sf I.7:\ll shifts \gg$  suggests a conversion of some wavelengths to some others. Maybe say  $\ll$  filters/absorbs longer wavelengths so that the downward irradiance spectrum is shifted towards shorter wavelengths  $\gg$

We reformulated this sentence to:

*"The transition from cloudy to cloudless atmospheric conditions increases the direct-toglobal ratio (fdir/glo) and the contribution of short wavelengths to the broadband downward irradiance (Warren, 1982)."*

I.9-10 : for which snowpack ?

We added this information in the revised manuscript:

*"For example, simulations with TARTES assuming cloudless and cloudy conditions changed the broadband snow surface albedo from about 0.8 to 0.9 for a SZA of 60" and a snow pack (no impurities) characterized by SSA= 20 m^2 kg^{-1}."*

**p.9**

Figure 2 :  $\ll$  adjusted  $\gg$  parameters is not clear. Do you mean that they can vary ? Maybe specify that the procedure is done at high spectral resolution so that the figure holds for a single wavelength. Consider adding a title with TARTES/libRadtran (or SNOW/ATMOSPHERE) on top of the colored boxes.

Thanks for the hint. We adjusted the scheme and figure caption: